# ❧ AC-REASON: Towards Theory-Guided Actual Causality Reasoning with Large Language Models

## Abstract

Actual causality (AC), a fundamental aspect of causal reasoning (CR), concerns attribution and responsibility assignment in real-world scenarios. However, existing LLM-based methods lack grounding in formal AC theory, resulting in limited interpretability. Therefore, we propose AC-REASON, a *semi-formal* reasoning framework that identifies causally relevant events within an AC scenario, infers the values of formal causal factors (e.g., sufficiency, necessity, and normality), and answers AC queries via a theory-guided algorithm with explanations. While AC-REASON does not explicitly construct a causal graph, it operates over variables in the underlying causal structure to support principled reasoning. To enable comprehensive evaluation, we introduce AC-BENCH, a new benchmark built upon and extending Big-Bench Hard Causal Judgment (BBH-CJ). AC-BENCH comprises ~1K carefully annotated samples, each with detailed reasoning steps and focuses solely on actual causation. The case study shows that synthesized samples in AC-BENCH present greater challenges for LLMs. Extensive experiments on BBH-CJ and AC-BENCH show that AC-REASON consistently improves LLM performance over baselines. On BBH-CJ, all tested LLMs surpass the average human accuracy of 69.60%, with GPT-4 + AC-REASON achieving 75.04%. On AC-BENCH, GPT-4 + AC-REASON again achieves the highest accuracy of 71.82%. Fine-grained analysis reveals with AC-REASON, LLMs exhibit more faithful reasoning, especially Qwen-2.5-72B-Instruct and Claude-3.5-Sonnet. Finally, our ablation study proves that integrating AC theory into LLMs is highly effective, with the proposed algorithm contributing the most significant performance gains.

## 1 Introduction

Causality is commonly divided into *type causality* and *actual causality* [31]. Type causality concerns variable-level relationships and effects within a causal structure, while actual causality (AC) concerns attribution and responsibility assignment in real-world applications such as legal reasoning [32] and legal responsibility [2]. An example of AC is illustrated in Figure 1. In the era of LLMs, studies mainly focus on eliciting their causal reasoning (CR) capabilities [26, 35, 25, 36, 34]. These efforts primarily address type causality tasks such as variable-level causal discovery and effect inference, and have achieved promising results [26, 35, 34]. However, AC remains relatively underexplored, and existing LLM-based methods targeting AC consistently fall short, often underperforming human-level accuracy [6]. For example, on Big-Bench Hard Causal Judgment (BBH-CJ) [50], GPT-4 with manually crafted chain-of-thought (CoT) exemplars achieves a state-of-the-art accuracy of 68.2%, still below the average human rater accuracy of 69.6% [51]. These methods also exhibit limited interpretability, as our analysis reveals that they often generate explanations that appear plausible but are theoretically incorrect. This motivates the central question of our work: *Can AC theory-guided LLMs perform more formal and accurate AC reasoning with theoretically aligned explanations?*

Submitted to 39th Conference on Neural Information Processing Systems (NeurIPS 2025). Do not distribute.

**Story:** Janet is an employee in a factory. Since she works in the maintenance department, she knows how to grease and oil all of the machines in the factory. [1] It is her responsibility to put oil into the machines. Kate is also an employee at the factory. While she works in the human resources department, she knows how to grease and oil all of the machines in the factory. [2] If Janet does not put oil in the machines, it is not Kate's responsibility to do so. [3] One day, Janet forgets to put oil in an important machine. [4] Kate noticed that Janet did not put oil in the machine, and Kate also did not put oil in the machine. The machine broke down a few days later.
**Question:** Did Kate not putting oil in the machine cause the machine to break down?
**Ground Truth Answer:** Yes

**Causal Setting Establishment**

```
{
    "causal_events": {
        "Kate does not put oil in the machine": {
            "occur": True,
            "order": 1,
            "focal": True
        },
        "Janet does not put oil in the machine": {
            "occur": True,
            "order": 0,
            "focal": True
        }
    },
    "outcome_event": {
        "The machine breaks down": {
            "occur": 1,
            "order": 2
        }
    }
}
```

**Causal Factors Analysis**

```
{
    "causal_events": {
        "Kate does not put oil in the machine": {
            "sufficient": False,
            "necessary": True,
            "halpern_pearl": True,
            "norm_violated": False,  # [2]
            "behavior_intended": True  # [4]
        },
        "Janet does not put oil in the machine": {
            "sufficient": False,
            "necessary": True,
            "halpern_pearl": True,
            "norm_violated": True,  # [1]
            "behavior_intended": False  # [3]
        }
    }
}
```

**Actual Causality Reasoning**

**Input:** the *structured causal knowledge* from causal factors analysis, including the *causal events* and their *inferred factor values*.

**Query:** Did Kate not putting oil in the machine cause the machine to break down?

**Answer:** Yes.

**Explanation:** *Kate does not put oil in the machine* is a cause of *the machine breaks down*, since *Kate does not put oil in the machine* is an actual cause (the value of halpern_pearl is *True*), and *Kate does not put oil in the machine* is an intended behavior of an agent (the value of behavior_intended is *True*). Based on Line 16 of Algorithm 1, *Kate does not put oil in the machine* is a judged cause of *the machine breaks down*.

Figure 1: A data example in AC-BENCH. Actual causality (AC) features individual-level causality between real-world events. To derive the actual causal relationships between these events, we need to consider 1) the outcomes and their candidate causes, 2) the factors that determine whether each candidate cause is at least part of a cause of the outcome, and 3) the factors that influence the responsibility of each candidate cause, such as normality and agent intent [47].

To address this, we introduce the AC-REASON framework, which integrates LLMs with AC theory-guided CoT to enable more formal and interpretable AC reasoning. The framework operates in three stages. First, it identifies causally relevant events—candidate causes and outcomes—from an actual causal scenario. Second, it infers the values of formal causal factors related to AC (e.g., sufficiency, necessity, and normality) for each candidate cause. Third, it determines actual causal relationships using a theory-grounded algorithm that reasons over the inferred factor values to answer AC queries (e.g., *Does X cause Y?*), along with a theoretically aligned explanation. **The Role of LLMs.** Formal causal models struggle to define certain background contexts that shape human *causal judgment* [11], including the causal frame (i.e., the set of candidate causes relevant to an outcome), sufficiency, necessity, normality, and epistemic state [30]. In our framework, LLMs act as *event detectors* that set the causal frame of a causal scenario after the outcome occurs, simulating human attribution. This reduces the need for expert intervention in manually defining the causal frame [31]. Second, LLMs act as *factor value reasoners*, as recent studies have shown their effectiveness in inferring the values of key causal factors for AC reasoning [31]. For instance, GPT-4 infers sufficient and necessary causes with an average accuracy over 80% [31]. **The Role of AC Theory.** AC theory plays two complementary roles in AC-REASON. First, it acts as a *factor definition provider* by translating formal definitions of causal factors into natural language descriptions for LLMs, such as the Halpern-Pearl definition [16]. Second, it acts as a *reasoning algorithm guider*, guiding the design of our algorithm that determines actual causal relationships given the inferred factor values.

In addition to methodological limitations, evaluation of AC reasoning has been largely overlooked in existing CR benchmarks for LLMs [26, 35, 38]. The most widely used dataset, BBH-CJ, includes only 187 samples and lacks fine-grained annotations of reasoning steps. To address this gap, we introduce AC-BENCH, an extension of BBH-CJ that enables more comprehensive evaluation of AC reasoning. AC-BENCH features detailed, manually annotated reasoning steps for each query and expands the data size leveraging the data synthesis capabilities of LLMs. It focuses solely on actual causation, and our case study reveals that LLM-generated samples present greater challenges derived from the generation pipeline. For example, when generating a new story, new details with spurious correlations may be introduced, and important but non-essential causal cues may be omitted.

The contributions are threefold: (1) We propose the AC-REASON framework, which—to the best of our knowledge—is the first to integrate LLMs with AC theory for more formal and interpretable

AC reasoning. (2) We construct the AC-BENCH benchmark, which focuses solely on actual causal relationships. It scales BBH-CJ to ~1K samples and incorporates more challenging samples, enabling more comprehensive evaluation. (3) Experiments show that AC-REASON consistently improves performance across various LLMs, outperforming other baselines. Also, the integration of AC theory into LLMs via our algorithm is proved to be highly effective by the ablation study.

## 2 Preliminaries

AC cares about the actual causal relationships between candidate causes and outcomes, referred to as *causal events* and *outcome events*, respectively. These correspond to the endogenous variables in an underlying causal structure. Given a natural language causal story $t$ from an AC problem, a *(recursive) causal model* $M$ is formally defined as a pair $(\mathcal{S}, \mathcal{F})$, where $\mathcal{S} = (\mathcal{U}, \mathcal{V}, \mathcal{R})$ is the *signature* consisting of a set of exogenous variables $\mathcal{U}$, a set of endogenous variables $\mathcal{V}$, and the *ranges* of variables in $\mathcal{V}$. $\mathcal{F}$ contains the *structural equations* that determine the values of endogenous variables. $\mathcal{V}$ is partitioned into the set of causal events $\mathcal{C}$ and the set of outcome events $\mathcal{O}$. Each event $E \in \mathcal{V}$ takes values from $\mathcal{R}(E) = \{0, 1\}$, indicating whether $E$ actually occurs.

An AC query $q$ asks whether a conjunction of causal events ($\wedge$) causes a combined outcome event ($\wedge$, $\vee$, or $\neg$). We present the causal events in conjunction as $\boldsymbol{X} = \boldsymbol{x}$, where each variable $X$ takes on the value $x$, and the combined outcome event as $\varphi$. A *causal formula* $\psi$ is written as $[\boldsymbol{Y} \leftarrow \boldsymbol{y}]\varphi, \boldsymbol{Y} \subset \mathcal{V}$, meaning that $\varphi$ would hold if an intervention set the variables in $\boldsymbol{Y}$ to $\boldsymbol{y}$. A *causal setting* is a pair $(M, \boldsymbol{u})$, where $M$ is a (recursive) causal model and $\boldsymbol{u}$ is a *context*—an assignment to the exogenous variables in $\mathcal{U}$. If a formula $\psi$ holds in the causal setting $(M, \boldsymbol{u})$, we write $(M, \boldsymbol{u}) \models \psi$.

### 2.1 The Halpern-Pearl Definition of Causality

The Halpern-Pearl (HP) definition of causality is a formal definition of a (partial) cause in the AC background [16]. Here, we present the *modified* version of the HP definition:

**Definition 1.** $\boldsymbol{X} = \boldsymbol{x}$ *is an actual cause of* $\varphi$ *in the causal setting* $(M, \boldsymbol{u})$ *if the following three conditions hold:*

    *AC1.* $(M, \boldsymbol{u}) \models (\boldsymbol{X} = \boldsymbol{x})$ *and* $(M, \boldsymbol{u}) \models \varphi$.

    *AC2. There exists a set* $\boldsymbol{W}$ *of variables in* $\mathcal{V}$ *and a setting* $\boldsymbol{x}'$ *of variables in* $\boldsymbol{X}$ *such that if* $(M, \boldsymbol{u}) \models \boldsymbol{W} = \boldsymbol{w}^*$, *then*

$$(M, \boldsymbol{u}) \models [\boldsymbol{X} \leftarrow \boldsymbol{x}', \boldsymbol{W} \leftarrow \boldsymbol{w}^*]\neg\varphi.$$

    *AC3.* $\boldsymbol{X}$ *is minimal; there is no strict subset* $\boldsymbol{X}'$ *of* $\boldsymbol{X}$ *such that* $\boldsymbol{X}' = \boldsymbol{x}'$ *satisfies conditions AC1 and AC2, where* $\boldsymbol{x}'$ *is the restriction of* $\boldsymbol{x}$ *to the variables in* $\boldsymbol{X}$.

AC1 requires that both $\boldsymbol{X} = \boldsymbol{x}$ and $\varphi$ actually occur in the causal setting $(M, \boldsymbol{u})$; AC2 establishes a permissive counterfactual dependence of $\varphi$ on $\boldsymbol{X}$ by forcing the variables in $\boldsymbol{W}$ fixed at their actual values; AC3 ensures minimality, requiring only essential components are included. Overall, the HP definition can be viewed as an extension of the but-for test: $\boldsymbol{X} = \boldsymbol{x}$ is a cause of $\varphi$ if, but for $\boldsymbol{X} = \boldsymbol{x}$, $\varphi$ would not have happened. Based on this, we state Proposition 1 [16], proved in Appendix A.3.

**Proposition 1.** *If* $\boldsymbol{X} = \boldsymbol{x}$ *is a but-for cause of* $\varphi$ *in the causal setting* $(M, \boldsymbol{u})$, *then* $\boldsymbol{X} = \boldsymbol{x}$ *is a cause of* $\varphi$ *according to the HP definition.*

According to Proposition 1, if $\boldsymbol{X} = \boldsymbol{x}$ is a but-for (i.e., necessary) cause of $\varphi$, then it also qualifies as an actual cause of $\varphi$. However, the notion of an *actual cause* as defined in Definition 1 corresponds only to *part of a cause* [16]. To ensure a more complete account of causality, additional factors such as sufficiency, normality, intention, and responsibility are necessary. For clarity, we treat *actual cause* as synonymous with *part of a cause*, while *cause* refers to a cause derived through judgment.

### 2.2 The Completeness of Actual Causes

One way to assess the completeness of an actual cause $\boldsymbol{X} = \boldsymbol{x}$ is to examine whether it also satisfies sufficiency for the occurrence of $\varphi$ [16]. Additional relevant factors are discussed in Appendix A.2.

**Definition 2.** $\boldsymbol{X} = \boldsymbol{x}$ is a sufficient cause of $\varphi$ in the causal setting $(M, \boldsymbol{u})$ if the following four conditions hold:

*SC1.* $(M, \boldsymbol{u}) \models (\boldsymbol{X} = \boldsymbol{x})$ and $(M, \boldsymbol{u}) \models \varphi$.

*SC2.* Some conjunct of $\boldsymbol{X} = \boldsymbol{x}$ is part of a cause of $\varphi$ in the causal setting $(M, \boldsymbol{u})$.

*SC3.* $(M, \boldsymbol{u}') \models [\boldsymbol{X} \leftarrow \boldsymbol{x}]\varphi$ for all contexts $\boldsymbol{u}'$.

*SC4.* $\boldsymbol{X}$ is minimal; there is no strict subset $\boldsymbol{X}'$ of $\boldsymbol{X}$ such that $\boldsymbol{X}' = \boldsymbol{x}'$ satisfies conditions SC1, SC2, and SC3, where $\boldsymbol{x}'$ is the restriction of $\boldsymbol{x}$ to the variables in $\boldsymbol{X}$.

SC1 and SC4 align with AC1 and AC3, respectively. SC2 requires that a sufficient cause must include at least an actual cause. SC3 demands that $\boldsymbol{X} = \boldsymbol{x}$ suffices to cause $\varphi$ across all contexts—that is, regardless of how the values of other causal events vary, $\boldsymbol{X} = \boldsymbol{x}$ consistently causes $\varphi$. If $\boldsymbol{X} = \boldsymbol{x}$ qualifies as a sufficient actual cause of $\varphi$, it can be reasonably concluded to be a cause of $\varphi$.

# 3 AC-REASON Framework

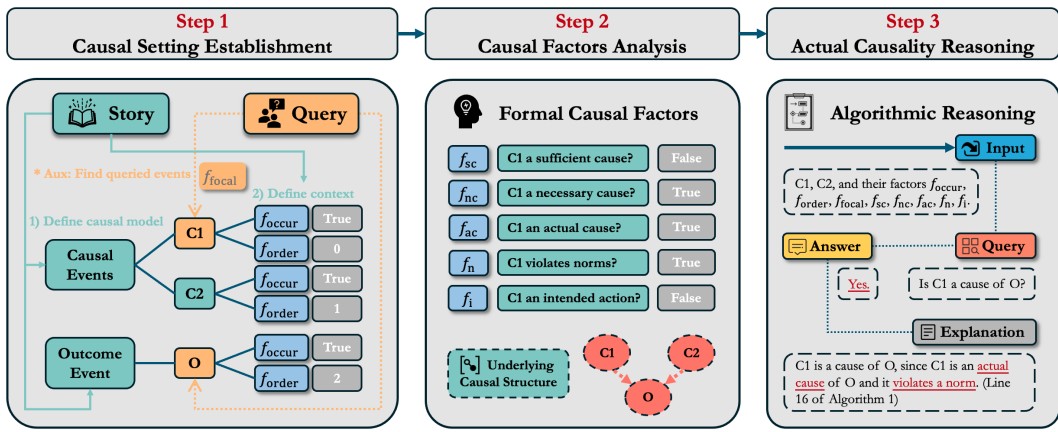

Figure 2: The overview of AC-REASON. The causal structure at the bottom is not actually constructed.

As illustrated in Figure 2, AC-REASON operates in three stages. First, it establishes the causal setting by extracting causally dependent events and the context from the story. Second, it infers the values of causal factors for causal events. Third, it performs AC reasoning via algorithmic reasoning, returns an answer and an explanation. The prompts for AC-REASON are presented in Appendix A.7.

## 3.1 Causal Setting Establishment

In this step, AC-REASON establishes the causal setting $(M, \boldsymbol{u})$.

**Causal Model.** First, AC-REASON instructs an LLM to extract causal events $\mathcal{C}$ and outcome events $\mathcal{O}$ from the story $t$, and combine the outcome events to a unified outcome event $O$. The instruction states and generally ensures that causal events causally contribute to the outcome event and are minimal and non-overlapping (AC3). Second, AC-REASON records whether each causal event is queried using a binary factor $f_{focal}$. If $E \in \mathcal{C}$ is queried (i.e., a *focal causal event*), $f_{focal}^E$ holds. In summary, AC-REASON identifies the set of causal events $\mathcal{C}$ and the unified outcome event $O$ to define the causal model $M$, and assigns an auxiliary factor $f_{focal}$.

**Context.** Formally, the context $\boldsymbol{u}$ consists of the values of exogenous variables in $\mathcal{U}$, which in turn determine the values of endogenous variables via the structural equations $\mathcal{F}$ [16]. In practice, AC-REASON extracts the values of endogenous variables—that is, whether each event occurs—as a proxy for the context. Additionally, it captures the temporal order of events to support downstream reasoning. For each event $E \in \mathcal{V}$, its occurrence is represented by a binary factor $f_{occur}$, where $f_{occur}^E$ holds if $E$ actually occurs. Its temporal order is recorded using an integer-valued factor $f_{order}$, where $f_{order}^E$ indicates the relative sequential position of $E$ starting from 0.

The step of causal setting establishment is formalized by the following formula:

$$(\mathcal{C}, O, f_{focal}, f_{occur}, f_{order}) \leftarrow \Phi(t, q, I_m, I_c),$$

where $\Phi$ is an LLM. $t$, $q$, $I_m$, and $I_c$ denote the story, query, and the instructions for extracting the causal model and context, respectively.

## 3.2 Causal Factors Analysis

In this step, AC-REASON infers the values of 5 binary causal factors for each causal event $E \in \mathcal{C}$.

Fac. 1. $f_{sc}$ determines whether $E$ is a sufficient cause of $O$ based on Definition 2. If had $E$ occurred, $O$ would have occurred under all contexts, $f_{sc}^E$ holds. Here, "under all contexts" means counterfactual reasoning over other causal events, ensuring robust sufficiency [22, 53].

Fac. 2. $f_{nc}$ determines whether $E$ is a necessary cause of $O$ based on the but-for definition. If had $E$ not occurred, $O$ would not have occurred, $f_{nc}^E$ holds. The counterfactual reasoning allows for the potential changes in other causal events when altering $E$.

Fac. 3. $f_{ac}$ determines whether $E$ is an actual cause based on Definition 1. If had $E$ not occurred, $O$ would not have occurred, while allowing at least one subset of causal events other than $E$ to remain unchanged when $E$ is altered, $f_{ac}$ holds. AC3 is satisfied, as $E$ is atomic (as discussed in Section 3.1). Furthermore, when $f_{nc}$ holds, $f_{ac}$ should also hold (Proposition 1).

Fac. 4. $f_n$ determines whether $E$ violates a descriptive or prescriptive norm.

Fac. 5. $f_i$ determines whether $E$ is an agent's action, and the agent is aware of the potential outcomes of their action and knowingly performs an action that leads to a foreseeable bad outcome.

The step of causal factors analysis is formalized by the following formula:

$$(f_{sc}, f_{nc}, f_{ac}, f_n, f_i) = \Phi(t, I_f, \mathcal{C}, O, f_{occur}),$$

where $I_f$ is the instruction for inferring the factor values. $f_{occur}$ is used for counterfactual reasoning.

## 3.3 Actual Causality Reasoning

Based on the extracted causal knowledge, we design Algorithm 1 for AC reasoning. The algorithm formalizes the use of such structured knowledge to answer AC queries with explanations. For each focal causal event $E$, the algorithm first partitions the reasoning process using $f_{sc}^E$, $f_{nc}^E$ and $f_{ac}^E$ (Lines 4, 5, 6, and 15). Then, for each partition, it determines the causal relationship between $E$ and $O$, drawing on established theory in actual causality and causal judgment. Finally, it outputs a binary answer to the query and provides an explanation. Specifically, 1) If $E$ is both sufficient and necessary, it is a cause of $O$ since it has a responsibility of 100% (Line 4). 2) If $E$ is neither sufficient nor necessary, it is not a cause of $O$ since it has a responsibility of 0% (Line 5). 3) If $E$ is one among multiple disjunctive causal events, the algorithm examines their temporal order to assess potential *preemption* [17, 16] (Line 9-10). If these events occur simultaneously, their respective responsibilities based on $f_n$ and $f_i$ are compared to determine the cause (Line 13-14). 4) If $E$ is part of a conjunctive set of causal events, the algorithm first evaluates whether $f_n^E$ and $f_i^E$ enhance $E$'s causal strength (Line 16). If so, $E$ is judged as a cause. Otherwise, their respective responsibilities based on $f_{order}$ are compared to determine the cause (Lines 20-21). Details on reasoning process partitioning, rule justification for each partition, and explanation generation are presented in Appendices A.3 and A.5.

# 4 AC-BENCH Construction

Like the example in Figure 1, AC-BENCH is defined as $\mathcal{D} := \{t_i, q_i, a_i, r_i\}_{i=1}^{N}$, where each sample comprises a causal story $t_i$, a causal query $q_i$, a binary answer $a_i \in \{\text{Yes}, \text{No}\}$, and the reasoning steps $r_i$. The goal of AC-BENCH is to evaluate the correctness of an LLM to map a causal query to a binary answer, and the reasoning steps. The pipeline for constructing AC-BENCH is detailed below.

**Data Cleaning.** The source of AC-BENCH is BBH-CJ [50], which contains 187 samples. Before annotation, we conduct thorough data cleaning on BBH-CJ by removing and correcting samples. This process reduces noise and preserves only samples of interest, resulting in a curated set of 133 samples. Further details and examples are provided in Appendix A.6.

**Algorithm 1:** Actual Causality Reasoning

**Input:** Causal events $\mathcal{C}$; factors $(f_{focal}^E, f_{occur}^E, f_{order}^E, f_{sc}^E, f_{nc}^E, f_{ac}^E, f_n^E, f_i^E)$ for each $E \in \mathcal{C}$.
**Output:** A Yes/No answer to the query; an explanation (discussed in Appendix A.5).

1   $\mathcal{A} \leftarrow \{\}$;
2   **for each** $E \in \mathcal{C}$ **do**
3     **if** $\neg f_{focal}^E$ **then** CONTINUE;
4     **if** $f_{sc}^E \wedge f_{nc}^E$ **then** $\mathcal{A}$.append("Yes");
5     **else if** $\neg f_{sc}^E \wedge \neg f_{ac}^E$ **then** $\mathcal{A}$.append("No");
6     **else if** $f_{sc}^E \wedge \neg f_{nc}^E$ **then**
7       $\mathcal{C}_s \leftarrow \{E' \in \mathcal{C} \mid f_{sc}^{E'} \wedge \neg f_{nc}^{E'}\}$;
8       **if** $\text{len}(\text{set}(\{f_{order}^{E_s} \mid E_s \in \mathcal{C}_s\})) \neq 1$ **then**
9         **if** $f_{order}^E = \max\{f_{order}^{E_s} \mid E_s \in \mathcal{C}_s\}$ **then** $\mathcal{A}$.append("Yes");
10        **else** $\mathcal{A}$.append("No");
11       **else**
12         **for each** $E_s \in \mathcal{C}_s$ **do** $f_{resp}^{E_s} \leftarrow \Phi(f_n^{E_s}, f_i^{E_s})$ ;
13         **if** $f_{resp}^E \leftarrow \max\{f_{resp}^{E_s} \mid E_s \in \mathcal{C}_s\}$ **then** $\mathcal{A}$.append("Yes");
14        **else** $\mathcal{A}$.append("No");
15     **else if** $\neg f_{sc}^E \wedge f_{ac}^E$ **then**
16       **if** $f_n^E \vee f_i^E$ **then** $\mathcal{A}$.append("Yes");
17       **else**
18         $\mathcal{C}_a \leftarrow \{E' \in \mathcal{C} \mid \neg f_{sc}^{E'} \wedge f_{ac}^{E'}\}$;
19         **for each** $E_a \in \mathcal{C}_a$ **do** $f_{resp}^{E_a} = \Phi(f_{order}^{E_a})$;
20         **if** $f_{resp}^E = \max\{f_{resp}^{E_a} \mid E_a \in \mathcal{C}_a\}$ **then** $\mathcal{A}$.append("Yes");
21        **else** $\mathcal{A}$.append("No");
22     **else** CONTINUE;
23   **return** "Yes" **if** $\mathcal{A}$ contains "Yes", **else** "No";

**Data Annotation.** We manually annotate the reasoning steps for each sample. Each annotation follows a structured template similar to the one shown in Figure 1, including the causal setting and the values of causal factors. The annotated information is organized into a unified JSON format. The annotation guidelines are detailed in Appendix A.6.

**Data Augmentation.** To expand the dataset, we manually create new samples from existing stories by altering the query to focus on different causal events. If a causal event $E \in \mathcal{C}$ is not queried, we generate a variant where $E$ becomes the focal causal event. In such cases, only $f_{focal}$ and $a$ need to be updated. This increases the data size to 163. To further scale the dataset, we employ GPT-4o [24] to synthesize new samples by rewriting stories from existing "seed samples". After generation, the dataset reaches 1093 samples. The generation prompts are presented in Appendix A.7.

**Data Verification.** We apply both automated validation and human evaluation to verify the quality of generated samples. The details are presented in Appendix A.6.

Table 1 summarizes the statistics of AC-BENCH. The dataset contains 1093 samples with nearly balanced positive and negative answers. On average, each reasoning involves 2.39 causal events and one outcome event. In rare cases, more than one focal causal event appears in a single sample.

## 5 Experiments

### 5.1 Setups and Baselines

In the pilot and main experiments, we evaluate AC-REASON using recent LLMs, including Qwen-2.5-32B/72B-Instruct [54], DeepSeek-V3 [33], Gemini-2.0-Flash [14], GPT-4-0613 [1], GPT-4o-2024-11-20 [24], and Claude-3.5-Sonnet [4]. For all models, the temperature is set to 0 (or approaches

| | Total |
|---|---|
| Table 1: AC-BENCH statistics. | |

| | Total |
|---|---|
| **Size** | |
| # Samples | 1093 |
| **Story** | |
| # Sentences/Sample | 8.49 |
| # Words/Sample | 162.70 |
| **Query** | |
| # Focals/Sample | 1.02 |
| **Reasoning** | |
| # Events/Sample | 3.39 |
| # Causes/Sample | 2.39 |
| # Outcomes/Sample | 1 |
| **Answer** | |
| Positive Class | 53.8% |

Table 2: Results of the pilot study.

| Methods | Acc. | Acc. (C.) | Acc. (I.) |
|---|---|---|---|
| Human Average [50] | 69.60% | - | - |
| Qwen-2.5-32B-Instruct | 68.98% | 65.25% | 80.43% |
| + AC-REASON | 72.55% | 69.98% | 80.43% |
| Qwen-2.5-72B-Instruct | 68.98% | 65.96% | 78.26% |
| + AC-REASON | 73.62% | 72.10% | 78.26% |
| DeepSeek-V3 | 69.70% | 67.14% | 77.54% |
| + AC-REASON | 73.44% | 71.39% | 79.71% |
| Claude-3.5-Sonnet | 66.49% | 65.01% | 71.01% |
| + AC-REASON | 74.33% | 73.29% | 77.54% |
| GPT-4o-2024-11-20 | 62.03% | 54.85% | 84.06% |
| + AC-REASON | 73.98% | 71.16% | 82.61% |
| GPT-4-0613 | 67.38% | 62.65% | 81.88% |
| + AC-REASON | **75.04%** | **74.70%** | 76.09% |

to 0) to ensure stable outputs. Additionally, we test AC-REASON with reasoning models such as
DeepSeek-R1 [15] and QwQ-32B [41] in the ablation study. The baselines are: 1) **Vanilla**, which
directly prompts the LLM to output a Yes/No answer given the story and query; 2) **Zero-shot CoT**,
which adds the instruction — *Let's think step by step* to the vanilla prompt; 3) **Manual CoT** [6],
which replaces *Let's think step by step* with three in-context examplars containing manually written
reasoning steps in natural language; 4) **AC-REASON**, our proposed multi-step CoT method guided
by formal AC theory. GPT-4 + manual CoT [6] represents the previous state-of-the-art on BBH-CJ.

## 5.2 Pilot Study

We begin with a pilot study on BBH-CJ to preliminarily evaluate the effectiveness of AC-REASON.
We report overall accuracy as well as fine-grained accuracies on two types of queries: causation
(C.) and intention (I.). As shown in Table 2, all models exhibit consistent improvements when using
AC-REASON and outperform the average accuracy of human raters. Notably, GPT-4 + AC-REASON
achieves an overall accuracy of 75.04%, significantly surpassing the human average of 69.60%.
Interestingly, closed-source LLMs benefit more from AC-REASON than open-source LLMs. We
attribute this to the stronger capacity of these LLMs to handle complex, multi-step CoT instructions.
The complete results of the pilot study and more conclusions are presented in Appendix A.4.1. Given
the promising results of AC-REASON on BBH-CJ, we proceed to conduct more detailed experiments
and analyses on our larger-scale dataset, AC-BENCH, to further validate its effectiveness.

## 5.3 Main Results

The main results are presented in
Table 3. Consistent with our pilot
study, all models improve consis-
tently equipped with AC-REASON,
and the performance gains are more
pronounced in closed-source LLMs.
This trend is especially evident in
GPT-4o, which achieves a remark-
able improvement of 9.42% with
AC-REASON. By comparing differ-
ent CoT prompting strategies, sev-
eral insights emerge: 1) Zero-shot
CoT fails to improve performance
across all models, highlighting the

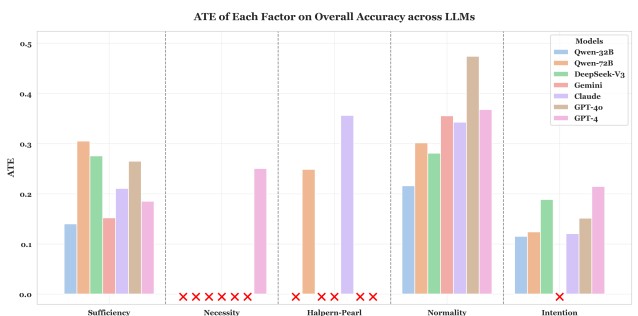

Figure 3: Results of the causal analysis.

inherent difficulty of the task. 2) Manual CoT—previously the state-of-the-art on BBH-CJ—yields
only marginal gains or even degrades performance in some cases. 3) In contrast, AC-REASON
delivers substantial improvements, demonstrating the benefits of grounding reasoning in formal
AC theory. To better understand the mechanism of AC-REASON, we further report fine-grained

Table 3: The main results of different LLMs on AC-BENCH. All results are presented in proportions. Apart from accuracy, we also present the fine-grained results of the reasoning steps. **CE✓** and **OE✓** stand for the proportions of correctly identified causal and outcome events, respectively.

| Methods | Accuracy | CE✓ | OE✓ | $f_{sc}$✓ | $f_{nc}$✓ | $f_{ac}$✓ | $f_n$✓ | $f_i$✓ |
|---|---|---|---|---|---|---|---|---|
| Random | 50.00% | | | | | | | |
| *Open-source LLMs* | | | | | | | | |
| Qwen-2.5-32B-Instruct | 62.67% | | | | | | | |
| + zero-shot CoT | 61.94%-0.73% | | | | | | | |
| + manual CoT | 62.85%+0.18% | | | | | | | |
| + AC-REASON | 64.78%+2.11% | 95.91% | 93.96% | 68.31% | 74.49% | 74.66% | 85.93% | 76.44% |
| Qwen-2.5-72B-Instruct | 64.87% | | | | | | | |
| + zero-shot CoT | 62.03%-2.84% | | | | | | | |
| + manual CoT | 62.31%-2.56% | | | | | | | |
| + AC-REASON | 67.52%+2.65% | 94.53% | 91.95% | 78.98% | **82.86%** | 74.82% | 86.04% | 78.71% |
| DeepSeek-V3 | 63.49% | | | | | | | |
| + zero-shot CoT | 64.68%+1.19% | | | | | | | |
| + manual CoT | 63.95%+0.46% | | | | | | | |
| + AC-REASON | 67.61%+2.93% | 96.06% | 91.40% | 84.41% | 60.38% | 64.28% | 87.04% | **80.42%** |
| *Closed-source LLMs* | | | | | | | | |
| Gemini-2.0-Flash | 60.20% | | | | | | | |
| + zero-shot CoT | 58.65%-1.55% | | | | | | | |
| + manual CoT | 58.55%-1.65% | | | | | | | |
| + AC-REASON | 64.96%+4.76% | **96.63%** | 95.61% | 67.15% | 69.28% | 72.78% | 88.31% | 74.15% |
| Claude-3.5-Sonnet | 63.68% | | | | | | | |
| + zero-shot CoT | 65.42%+1.74% | | | | | | | |
| + manual CoT | 62.67%+1.28% | | | | | | | |
| + AC-REASON | 70.54%+6.86% | 95.52% | 91.67% | 80.79% | 74.57% | 75.92% | **88.91%** | 74.12% |
| GPT-4o-2024-11-20 | 58.65% | | | | | | | |
| + zero-shot CoT | 59.38%+0.73% | | | | | | | |
| + manual CoT | 60.66%+2.01% | | | | | | | |
| + AC-REASON | 68.07%+9.42% | 94.45% | 92.04% | 80.92% | 79.39% | **78.04%** | 87.31% | 75.07% |
| GPT-4-0613 | 63.77% | | | | | | | |
| + zero-shot CoT | 62.49% -1.28% | | | | | | | |
| + manual CoT | 66.51%+2.74% | | | | | | | |
| + AC-REASON | **71.82%**+8.05% | 94.26% | **96.07%** | 74.30% | 72.87% | 72.96% | 87.69% | 74.57% |

accuracies for establishing the causal setting and inferring the values of causal factors. Interestingly, no obvious correlation is observed between these fine-grained accuracies and the overall accuracy. Therefore, we further conduct a factor analysis and a causal analysis in Appendix A.4.2, examining the extent to which each causal factor (causally) influences the overall performance. For the causal analysis, we report the average treatment effect (ATE) and standard error (SE) of each factor to the overall accuracy from two estimation methods—ordinary least squares (OLS) [3] and propensity score matching (Matching) [44]—across different LLMs in Table 6. In Figure 3, we illustrate the average ATE over OLS and Matching where $p < 0.05$, otherwise marked using cross marks. Ideally, all causal factors except $f_{nc}$—which is primarily used for partitioning and does not involve in determining causes—should causally contribute to the overall accuracy. However, only Qwen-2.5-72B-Instruct and Claude-3.5-Sonnet align well with this expectation, exhibiting faithful reasoning. GPT-4 appears to rely heavily on $f_{nc}$ while underutilizing $f_{ac}$, suggesting that it may be exploiting shortcuts rather than genuinely understanding the task. More broadly, the limited number of LLMs recognizing the importance of $f_{ac}$ underscores a common deficiency in current LLMs' grasp of actual causality.

## 5.4 Ablation Study

To evaluate the contribution of each stage in AC-REASON, we conduct an ablation study, with results summarized in Table 4. Due to the sequential dependency among the steps, we report performance for the following configurations: 1) **vanilla**; 2) **S1**, i.e., using only the first stage of AC-REASON; 3) **S12**, i.e., using the first two stages; and 4) AC-REASON. The prompts are presented in Appendix A.7. The results reveal three key findings: 1) Using only the first stage (S1) generally degrades performance across most LLMs. Although LLMs can identify relevant events independently (see Appendix A.4.3), the explicit prompting in S1 may lead them to extract superfluous or irrelevant

Table 4: Ablation results.

| Model | Vanilla | S1 | S12 | AC-REASON |
|---|---|---|---|---|
| *Open-source LLMs* | | | | |
| Qwen-2.5-32B-Instruct | 62.67% | 61.67%-1.00% | 64.59%+1.92% | 64.78%+2.11% |
| Qwen-2.5-72B-Instruct | 64.87% | 64.41%-0.46% | 67.98%+3.11% | 67.52%+2.65% |
| DeepSeek-V3 | 63.49% | 67.25%+3.76% | 65.87%+2.38% | 67.61%+4.12% |
| QwQ-32B | 54.99% | 56.17%+1.18% | 56.08%+1.09% | 67.15%+12.16% |
| DeepSeek-R1 | 57.09% | 58.28%+1.19% | 58.92%+1.83% | 69.72%+12.63% |
| *Closed-source LLMs* | | | | |
| Gemini-2.0-Flash | 60.20% | 58.19%-2.01% | 60.84%+0.64% | 64.96%+4.76% |
| Claude-3.5-Sonnet | 63.68% | 58.83%-4.85% | 60.93%-2.76% | 70.54%+6.86% |
| GPT-4o-2024-11-20 | 58.65% | 58.28%-0.37% | 58.46%-0.19% | 68.07%+9.42% |
| GPT-4-0613 | 63.77% | 62.31%-1.46% | 64.32%+0.55% | 71.82%+8.05% |

causal events, which ultimately hinders accurate decision-making. 2) Combining the first two stages (S12) typically improves performance. By guiding LLMs to consider distinct causal factors for each event, this step forsters a clearer understanding of the causal structure and facilitates more informed reasoning. 3) Further incorporating the third stage (S123) leads to the most substantial performance gains, validating our central hypothesis: the proposed Algorithm 1 enables precise AC reasoning by leveraging structured causal knowledge, consistent with the principles of AC theory. Given that the LLMs used in the main experiment generally lack intrinsic reflection or verification capabilities and struggle to accurately infer the values of causal factors, we further evaluate AC-REASON on "slow thinking" models such as DeepSeek-R1 [15] and QwQ-32B [41]. As shown in Table 4, although these models possess mechanisms for reflection and verification, they still perform poorly independently, highlighting the inherent complexity of AC reasoning. Nevertheless, they benefit from each individual stage of AC-REASON, achieving the highest performance gains among all tested models. This suggests that reflection and verification are effective only when well-instructed.

# 6  Related Work

Recent studies focus on evaluating the causal reasoning (CR) capabilities of LLMs [26, 27, 35, 6, 51, 38] and leveraging them for (*formal*) CR [25, 36]. In terms of datasets, evaluations have been developed to assess both the overall [6] and fine-grained [26] CR abilities of LLMs. On the methodological side, existing works typically enhance CR through strategies such as chain-of-though (CoT) prompting [50, 26], fine-tuning [25], or by integrating external tools [37]. These efforts collectively highlight the promise of LLMs in CR. However, actual causality (AC) reasoning remains relatively underexplored. Currently, the only widely used benchmark in the era of LLMs is Big-Bench Hard Causal Judgment [50]. Only a few works aim to improve the AC reasoning capabilities of LLMs [31, 6]. These either focus on analyzing isolated elements related to AC without addressing high-level AC queries [31], or apply prompting strategies such as CoT and in-context learning (ICL) [6]. As such, they fall short of fully engaging with the theoretical underpinnings of AC and offer limited interpretability. To bridge this gap, we propose AC-REASON and introduce AC-BENCH to foster more rigorous and interpretable research into the AC reasoning capabilities of LLMs.

# 7  Conclusion

In this work, we propose the AC-REASON framework, which—to the best of our knowledge—is the first to integrate LLMs with AC theory for more formal and interpretable AC reasoning. Also, we construct the AC-BENCH benchmark, which focuses solely on actual causal relationships. It scales BBH-CJ to ~1K samples and incorporates more challenging samples, enabling more comprehensive evaluation. Experiments on BBH-CJ and AC-BENCH show that AC-REASON consistently improves LLM performance over baselines. On BBH-CJ, all tested LLMs surpass the average human accuracy of 69.60%, with GPT-4 + AC-REASON achieving 75.04%. On AC-BENCH, GPT-4 + AC-REASON again achieves the highest accuracy of 71.82%. Finally, our ablation study proves that integrating AC theory into LLMs is highly effective, with the proposed algorithm contributing the most significant performance gains.

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

# A  Appendix

## A.1  Open Access to Code & Data

The code and data of this paper are provided in the following anonymous Github repository: `https://anonymous.4open.science/r/ac_reason-720D/`.

## A.2  Background

### A.2.1  The Categories of Causality

Causality is commonly divided into *type causality* and *actual causality* [31]. Type causality (or general causality) concerns general causal relationships and effects between variables within an underlying causal structure. In contrast, actual causality (or token/specific causality) evaluates the extent to which particular events cause other specific events [16, 18]. For example, the question *Does smoking causes lung cancer?* pertains to type causality, while *Was Fred's smoking habit responsible for his lung cancer?* pertains to actual causality.

### A.2.2  Causal Factors in Actual Causality and Causal Judgment

The key factors in actual causality are various definitions of a "cause", such as necessary causality, sufficient causality, and the Halpern-Pearl (HP) definition of causality [16]. A necessary cause is defined according to the but-for test; the definition of a sufficient cause is provided in Definition 2; and the HP definition is presented in Definition 1. In addition, *responsibility* is also an important concept. Responsibility is inherently relative. In formal AC models, responsibility is often defined in a quasi-probabilistic manner: the naive responsibility of a causal event $C$ for an outcome event $O$ is given by $1/(1 + k)$, where $k$ is the minimal number of changes required in the occurrences of other causal events to make $C$ a necessary cause of $O$ [16]. However, in natural language contexts, responsibility is actually a multi-factorial and qualitative indicator of causal contribution. Therefore, in practical use, it can be defined as the relative degree to which a causal event contributes causally to an outcome event, in comparison to other contributing events below [31]. Lastly, *normality* is another relevant concept that overlaps both actual causality and causal judgment. In this paper, we focus on its role in the domain of causal judgment. In causal judgment, the relevant factors have been summarized by domain experts [40].

**Causal Structure.**  The *conjunctive* or *disjunctive* nature of a causal structure plays a critical role in causal judgment [52, 39, 47]. A conjunctive causal structure implies that the outcome occurrs only if all contributing events jointly occur, whereas a disjunctive structure indicates that any single event is sufficient to bring about the outcome.

**Normality.**  Normality is crucial for both actual causality and causal judgment. The terms *norm*, *normal*, and *normality* are inherently ambiguous. Normality can be interpreted both descriptively and prescriptively [16]. Descriptive norms refer to what is statistically typical—such as the mode, mean, or values close to them, while prescriptive norms refer to moral standards, legal rules, institutional policies, or even proper functioning in machines or biological systems [16]. Humans tend to ascribe more causality to abnormal events than to normal ones [28, 23].

**Epistemic State.**  The intent of an agent is also important for causal judgment. It is considered as part of their epistemic state, which typically concerns the agent's awareness when performing specific actions. An agent who is aware of the potential consequences of their action and deliberately performs an action that leads to a foreseeable bad outcome is judged more harshly than one who lacks the relevant knowledge. This distinction is often framed as knowledge vs. ignorance [45, 29].

**Action/Omission.**  People tend to identify actions, rather than omissions, as the causes of outcomes— a phenomenon known as "omission bias" [43, 5]. In scenarios whether one individual acts while another one doesn't, the one who acts is more likely to be judged as a cause [20, 7, 9, 13].

**Temporal Effect.**  Causal judgment is also influenced by the temporal order of events [12]. When multiple events unfold over time and lead to an outcome, people generally tend to identify later events, rather than earlier ones, as the actual cause [42]. However, this preference depends on how the events are causally related to one another [21, 48]. When earlier events determine the course of subsequent actions, they are more likely to be selected as causes [19].

In our formulation, we consider actual causality factors $f_{sc}$, $f_{nc}$, $f_{ac}$, and $f_{resp}$. The first three represent different definitions of causality, while the last one is essential for assigning responsibility. Additionally, we include the factor $f_{occur}$ to ensure the satisfaction of AC1 in Definition 1. For causal judgment factors, we consider $f_n$ and $f_i$, corresponding to normality and the epistemic state, respectively. The temporal order of events is modeled using the factor $f_{order}$. It is worth noting that the notion of a "causal structure" implicitly corresponds to the factors $f_{sc}$ and $f_{nc}$: a conjunctive structure implies necessary causes, while a disjunctive structure implies sufficient causes. All causal factors incorporated in AC-REASON are summarized in Table 7.

## A.3 Theoretical Proofs

### A.3.1 Proof of Proposition 1

**Proposition 2.** *If $X = x$ is a but-for cause of $\varphi$ in the causal setting $(M, u)$, then $X = x$ is a cause of $\varphi$ according to the HP definition.*

*Proof.* Suppose $X = x$ is a but-for cause of $\varphi$. Then there exists at least a possible value $x'$ such that $(M, u) \models [X \leftarrow x'] \neg \varphi$. Let $W = \emptyset$ and consider the intervention $X \leftarrow x'$. Then AC2 is satisfied. Therefore, $X = x$ qualifies as an actual cause, as a special case where $W = \emptyset$. The proposition is proved. $\square$

### A.3.2 Proof of Algorithm 1

Formally, we consider the following factors in causal judgment:

$$(f_{occur}, f_{order}, f_{sc}, f_{nc}, f_{ac}, f_n, f_i, f_{resp}).$$

Let $E$ be a focal causal event and $O$ be the outcome event. We define the following:

- $f_{sc}^E$: Whether $E$ is part of a conjunctive structure, i.e., a necessary cause of $O$.
- $f_{nc}^E$: Whether $E$ is part of a disjunctive structure, i.e., a sufficient cause of $O$.
- $f_{ac}^E$: Whether $E$ is an actual cause of $O$ (based on Definition 1).
- $f_n^E$: Whether $E$ violates a prescriptive or descriptive norm.
- $f_i^E$: Whether $E$ is an agent's action, and the agent is aware of the potential outcomes of their action and knowingly performs an action that leads to a foreseeable bad outcome.
- $f_{order}^E$: The temporal order of $E$ relative to other events in the causal setting. It is an integer starting from 0, where simultaneous events share the same $f_{order}$.
- $f_{occur}^E$: Whether $E$ actually occurs in the causal setting.
- $f_{resp}^E$: The responsibility of $E$ relative to other causal events specified. It has different definitions in different settings.

First, for reasoning process partitioning (Lines 4, 5, 6, and 15), we use factors $(f_{sc}^E, f_{nc}^E, f_{ac}^E)$. When $E$ is sufficient, the partitioning is based on $f_{nc}^E$; when $E$ is not sufficient, the partitioning is based on $f_{ac}^E$. The objective of partitioning is to ensure that, within each partition, the causal relationship between $E$ and $O$ can be assessed using existing techniques from actual causality and causal judgment.

**Justification of Reasoning Process Partitioning**

**Case 1:** $f_{sc}^E$ holds (i.e., $E$ is sufficient). The partitioning is based on $f_{nc}^E$.

*Proof.* If $f_{nc}^E$ holds, $f_{ac}^E$ holds by Proposition 1. Since $f_{sc}^E$ also holds, $E$ has 100% responsibility for $O$, and thus the causal relationship is determined. If $\neg f_{nc}^E$ holds, the setting corresponds to cases such as preemption [17, 16] or overdetermination [46]. These are well-studied in literature, and they are defined primarily based on $f_{nc}$ rather than $f_{ac}$, validating this partition choice. $\square$

**Case 2:** $\neg f_{sc}^E$ holds (i.e., $E$ is not sufficient). The partitioning is based on $f_{ac}^E$.

*Proof.* If $f_{ac}^E$ holds, $E$ is at least part of a cause by Definition 1. We then only need to consider factors that increase causal strength to assess the causal relationship. If $\neg f_{ac}^E$ holds, $\neg f_{nc}^E$ holds by the contrapositive of Proposition 1. Since $\neg f_{sc}^E$ also holds, $E$ has 0% responsibility for $O$, and thus the causal relationship is determined. $\qquad\square$

The partitions are exhaustive, as they jointly cover the entire space of possibilities. Furthermore, within each partition, the causal relationship between $E$ and $O$ can be effectively assessed using existing techniques from actual causality and causal judgment. Therefore, the proposed reasoning process partitioning is well-founded. In the following, we will justify how to assess the causal relationship within each partition.

**Justification of Rules within Each Partition**

**Partition 1 (Line 4)**

*Proof.* $E$ is necessary, thus it is also an actual cause (Proposition 1). Therefore, $E$ is an actual cause that is also sufficient (100% responsibility), thus a cause of $O$.

$$f_{sc}^E \wedge f_{nc}^E \Rightarrow f_{sc}^E \wedge f_{ac}^E \Rightarrow E \text{ has 100\% responsibility for } O \Rightarrow E \text{ is a cause of } O.$$

$\qquad\square$

**Partition 2 (Line 5)**

*Proof.* $E$ is not an actual cause, thus it is also unnecessary (contrapositive of Proposition 1). Therefore, $E$ is insufficient and unnecessary (0% responsibility), thus not a cause of $O$.

$$\neg f_{sc}^E \wedge \neg f_{ac}^E \Rightarrow \neg f_{sc}^E \wedge \neg f_{nc}^E \Rightarrow E \text{ has 0\% responsibility for } O \Rightarrow E \text{ is not a cause of } O.$$

$\qquad\square$

**Partition 3 (Lines 6-14)**

*Proof.* $E$ is sufficient but not necessary, indicating the existence of a set of alternative sufficient causal events $\mathcal{C}_s$, each of which can independently cause $O$. In actual causality, typically only one causal event actually takes effect [16]. If events in $\mathcal{C}_s$ do not occur simultaneously, the one that occurs first is judged as the preemptive cause of $O$ (Line 9), while the others are not (Line 10) [17, 16]. Otherwise, if events in $\mathcal{C}_s$ occur simultaneously, their responsibilities should be compared. Here, responsibility is defined as the relative degree to which $E$ causally contributes to $O$ compared with other events in $\mathcal{C}s$. Causal contribution is evaluated using the remaining factors, i.e., $f_{resp}^E \leftarrow \Phi(f_{order}^E, f_n^E, f_i^E)$. If $E$ has the highest responsibility for $O$, it is a judged cause of $O$ (Line 13) since it actually takes effect; otherwise, it is not (Line 14). In the special case where all events in $\mathcal{C}_s$ have equal responsibility, the scenario corresponds to overdetermination [46], where each event in $\mathcal{C}_s$ is a judged cause of $O$. Formally, let $\mathcal{C}_s$ be the set of causal events that satisfy $f_{sc} \wedge \neg f_{nc}$ (including $E$), then:

$$\exists E_s \in \mathcal{C}_s, f_{order}^{E_s} < f_{order}^E \Rightarrow E_s \text{ preempts } E$$
$$\Rightarrow E \text{ does not actually take effect}$$
$$\Rightarrow E \text{ is not a judged cause of } O.$$

$$\forall E_s \in \mathcal{C}_s, f_{order}^{E_s} \geq f_{order}^E \Rightarrow E \text{ preempts all other events in } \mathcal{C}_s$$
$$\Rightarrow E \text{ actually takes effect}$$
$$\Rightarrow E \text{ is a judged cause of } O.$$

$$\forall E_s \in \mathcal{C}_s, f_{order}^{E_s} = f_{order}^E \wedge f_{resp}^E = \max_{E_s \in \mathcal{C}_s} f_{resp}^{E_s} \Rightarrow E \text{ actually takes effect}$$
$$\Rightarrow E \text{ is a judged cause of } O.$$

$$\forall E_s \in \mathcal{C}_s, f_{order}^{E_s} = f_{order}^E \wedge f_{resp}^E \neq \max_{E_s \in \mathcal{C}_s} f_{resp}^{E_s} \Rightarrow E \text{ does not actually take effect}$$
$$\Rightarrow E \text{ is not a judged cause of } O.$$

$\qquad\square$

**Partition 4 (Lines 15-21)**

*Proof.* Since $E$ is an actual cause, it is at least part of a cause by Definition 1. To determine whether $E$ is a judged cause of $O$, we consider factors that enhance causal strength. If a causal event involves factors that enhance its causal strength—though not sufficient on its own—it is still judged as a cause, as the judgment is binary. Factors that enhance causal strength include: 1) Normality. When $E$ violates norms, it is typically judged to be more of a cause of $O$ [28, 23]. 2) Intention. When $E$ is an agent's behavior that is intentional, and $O$ is adverse and foreseeable, it is typically judged to be more of a cause of $O$ [45, 29]. If $E$ satisfies either of these conditions, it is a judged cause of $O$ in AC-REASON (Line 16). If neither condition is met, we turn to its responsibility $f_{resp}^E \leftarrow \Phi(f_{order}^E)$, since $f_{order}^E$ remains the only unused factor and temporal order is known to influence causal judgment [12]. The principle is that, When earlier events determine the course of subsequent events, they are more likely to be judged as cause [19]. If $E$ is uniquely the most responsible for $O$, it is a judged cause of $O$ (Line 20); otherwise, it is not (Line 21). Formally, let $\mathcal{C}_a$ be the set of causal events that satisfy $\neg f_{sc} \wedge f_{ac}^E$ (including $E$), then:

$$f_{ac}^E \wedge (f_n^E \vee f_i^E) \Rightarrow E \text{ actually takes effect} \Rightarrow E \text{ is a judged cause of } O$$

$$f_{ac}^E \wedge \neg(f_n^E \vee f_i^E) \wedge \forall E_a \in \mathcal{C}_a, f_{resp}^E > f_{resp}^{E_a} \Rightarrow \text{The causal strength of } E \text{ is enhanced}$$
$$\Rightarrow E \text{ is not a judged cause of } O$$

$$f_{ac}^E \wedge \neg(f_n^E \vee f_i^E) \wedge \exists E_a \in \mathcal{C}_a, f_{resp}^E \leq f_{resp}^{E_a} \Rightarrow \text{The causal strength of } E \text{ is not enhanced}$$
$$\Rightarrow E \text{ is still not a sufficient cause}$$
$$\Rightarrow E \text{ is not a judged cause of } O$$

$\square$

## A.4 Empirical Results

Table 5: Complete results of the pilot study.

| Methods | Acc. | Acc. (C.) | Acc. (I.) |
|---|---|---|---|
| Human Average | 69.60% | - | - |
| Qwen-2.5-32B-Instruct | 68.98%±0.00% | 65.25%±0.00% | 80.43%±0.00% |
| + zero-shot CoT | 65.24%±0.44% | 61.94%±0.67% | 75.36%±1.02% |
| + manual CoT | 69.34%±0.25% | 66.19%±0.33% | 78.99%±1.02% |
| + AC-REASON | 72.55%±1.10% | 69.98%±1.46% | 80.43%±0.00% |
| Qwen-2.5-72B-Instruct | 68.98%±0.00% | 65.96%±0.00% | 78.26%±0.00% |
| + zero-shot CoT | 65.60%±1.26% | 61.23%±1.86% | 78.99%±1.02% |
| + manual CoT | 66.67%±0.91% | 62.65%±1.34% | 78.99%±1.02% |
| + AC-REASON | 73.62%±0.50% | 72.10%±0.67% | 78.26%±0.00% |
| DeepSeek-V3 | 69.70%±1.82% | 67.14%±1.77% | 77.54%±2.05% |
| + zero-shot CoT | 71.48%±0.50% | 70.69%±1.77% | 73.91%±3.55% |
| + manual CoT | 66.13%±0.91% | 62.65%±1.46% | 76.81%±1.02% |
| + AC-REASON | 73.44%±0.67% | 71.39%±0.67% | 79.71%±1.02% |
| GPT-4-0613 | 67.38%±0.76% | 62.65%±0.33% | 81.88%±2.05% |
| + zero-shot CoT | 67.74%±0.25% | 63.12%±0.58% | 81.88%±1.02% |
| + manual CoT | 68.81%±0.67% | 65.72%±0.88% | 78.26%±0.00% |
| + AC-REASON | **75.04%**±0.67% | **74.70%**±0.67% | 76.09%±1.77% |
| GPT-4o-2024-11-20 | 62.03%±0.44% | 54.85%±0.33% | 84.06%±1.02% |
| + zero-shot CoT | 65.42%±0.25% | 61.23%±0.33% | 78.26%±0.00% |
| + manual CoT | 64.88%±0.50% | 59.57%±0.58% | 81.16%±1.02% |
| + AC-REASON | 73.98%±0.67% | 71.16%±0.88% | **82.61%**±0.00% |
| Claude-3.5-Sonnet | 66.49%±0.91% | 65.01%±1.46% | 71.01%±5.42% |
| + zero-shot CoT | 62.03%±1.16% | 60.05%±2.34% | 68.12%±4.10% |
| + manual CoT | 68.98%±0.44% | 64.07%±0.88% | 84.06%±2.71% |
| + AC-REASON | 74.33%±1.51% | 73.29%±0.88% | 77.54%±4.47% |

### A.4.1 Pilot Experiment: Complete Results and Further Conclusions

The complete results of the pilot study is shown in Table 5. Specifically, we report both overall and fine-grained accuracies along with standard deviations over three runs. The results include 3 open-source and 3 closed-source LLMs under 4 different settings, providing comprehensive experimental coverage. In BBH-CJ [50], there are a total of 187 queries, comprising 141 causation queries and 46 intention queries. Based on these results, we highlight the following key findings. First, for causation queries, all models exhibit consistent performance improvements with the integration of AC-REASON, with open-source LLMs generally benefiting more. For instance, GPT-4's accuracy on causation queries increase from 62.65% to 74.70% after applying AC-REASON. GPT-4 + AC-REASON also achieves the highest overall accuracy of 75.04%, significantly surpassing the average human performance of 69.60%. These results suggest that our work offers a promising direction for actual causal reasoning in the era of LLMs—namely, incorporating domain theory into LLMs to enhance reasoning capabilities. Second, neither zero-shot nor manual CoT consistently improves model performance on causation queries. Zero-shot CoT only benefits GPT-4o and DeepSeek-V3 while manual CoT shows improvements exclusively for the GPT-series.

### A.4.2 Main Experiment: From Causal Factors to Overall Accuracy

Given the fine-grained results obtained in the main experiment, we explore the following question: *How does each causal factor (causally) contribute to the overall accuracy?* This analysis is essential because the individual accuracy of a factor merely reflects an LLM's understanding of that specific factor. It does not reveal how the predicted values of these factors (causally) influence the final outcome. Since AC-REASON employs Algorithm 1 rather than LLM-based reasoning to derive actual causality, this analysis serves as a "simulation" that helps us understand the relative importance of different causal factors for each LLM.

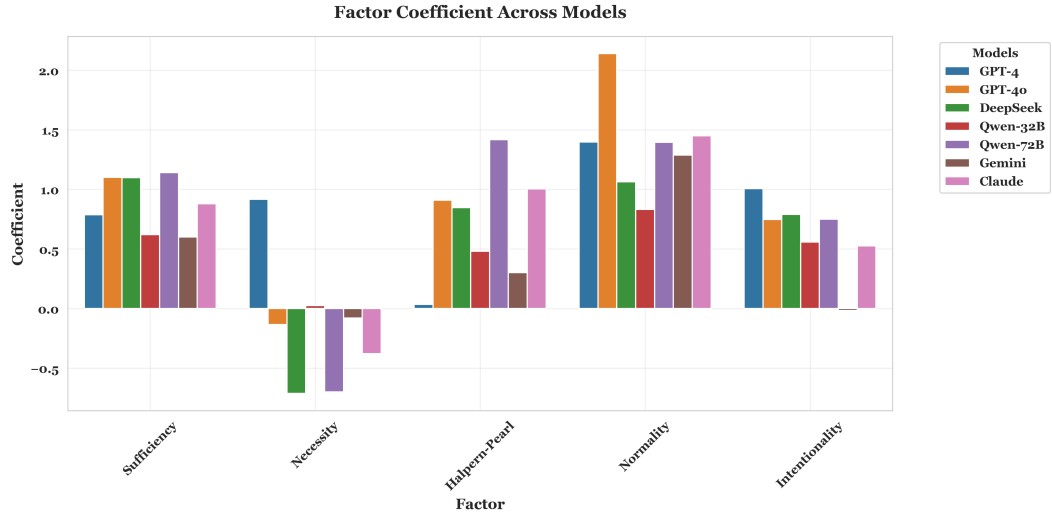

Figure 4: Results of the factor analysis.

To this end, we first conduct a *factor analysis* [10, 8] to examine the extent to which improving the accuracy of each individual factor increases the overall accuracy across different LLMs, with results reported in Figure 4. First, all LLMs consistently identify $f_{sc}$ and $f_n$ as important contributors to performance, while most also regard $f_{ac}$ and $f_i$ as influential. These findings align with our expectations, as Algorithm 1 implicitly assumes all factors (except for $f_{nc}$) play a role in causal judgment. Interestingly, most LLMs consider $f_{nc}$ unimportant or even detrimental. This is unsurprising: $f_{nc}$ does not actually participate in determining causality. Instead, $f_{nc}$ only functions as a sufficient condition for $f_{ac}$ and plays a role in partitioning, but not in the final judgment. A notable outlier is GPT-4, which appears to rely heavily on naive counterfactual dependence ($f_{nc}$) rather than the formal notion of actual causality ($f_{ac}$). This suggests that, absent Algorithm 1, GPT-4 might produce lower accuracy due to its dependence on an informal causal heuristic. This interpretation is supported by results from our ablation study. Moreover, $f_i$ appears to have negligible effect for Gemini. We

hypothesize that this is because, when Gemini correctly predicts $f_i$, Algorithm 1 often does not enter the relevant decision branch (Line 16), rendering $f_i$ irrelevant to the final prediction in those cases.

Table 6: Results of the causal analysis.

| Model | Factor | OLS | | | Matching | | |
|---|---|---|---|---|---|---|---|
| | | ATE | SE | $p$-value | ATE | SE | $p$-value |
| Qwen-2.5-32B-Instruct | $f_{sc}$ | 0.141 | 0.031 | 0.000 | 0.139 | 0.034 | 0.000 |
| | $f_{nc}$ | -0.018 | 0.043 | 0.682 | -0.098 | 0.056 | 0.079 |
| | $f_{ac}$ | 0.068 | 0.045 | 0.130 | 0.083 | 0.063 | 0.189 |
| | $f_n$ | 0.227 | 0.040 | 0.000 | 0.205 | 0.049 | 0.000 |
| | $f_i$ | 0.120 | 0.033 | 0.000 | 0.110 | 0.037 | 0.003 |
| Qwen-2.5-72B-Instruct | $f_{sc}$ | 0.299 | 0.038 | 0.000 | 0.311 | 0.053 | 0.000 |
| | $f_{nc}$ | 0.027 | 0.164 | 0.868 | 0.131 | 0.144 | 0.360 |
| | $f_{ac}$ | 0.256 | 0.059 | 0.000 | 0.241 | 0.067 | 0.000 |
| | $f_n$ | 0.317 | 0.039 | 0.000 | 0.286 | 0.062 | 0.000 |
| | $f_i$ | 0.129 | 0.033 | 0.000 | 0.119 | 0.037 | 0.000 |
| DeepSeek-V3 | $f_{sc}$ | 0.296 | 0.052 | 0.000 | 0.255 | 0.061 | 0.000 |
| | $f_{nc}$ | -0.126 | 0.060 | 0.037 | 0.001 | 0.078 | 0.989 |
| | $f_{ac}$ | 0.095 | 0.088 | 0.280 | 0.016 | 0.089 | 0.858 |
| | $f_n$ | 0.284 | 0.043 | 0.000 | 0.278 | 0.058 | 0.000 |
| | $f_i$ | 0.197 | 0.032 | 0.000 | 0.180 | 0.040 | 0.000 |
| Gemini-2.0-Flash | $f_{sc}$ | 0.158 | 0.031 | 0.000 | 0.146 | 0.035 | 0.000 |
| | $f_{nc}$ | -0.005 | 0.037 | 0.890 | -0.004 | 0.041 | 0.915 |
| | $f_{ac}$ | 0.062 | 0.047 | 0.182 | 0.122 | 0.044 | 0.006 |
| | $f_n$ | 0.362 | 0.043 | 0.000 | 0.349 | 0.060 | 0.000 |
| | $f_i$ | 0.032 | 0.031 | 0.305 | 0.043 | 0.036 | 0.228 |
| Claude-3.5-Sonnet | $f_{sc}$ | 0.212 | 0.036 | 0.000 | 0.209 | 0.041 | 0.000 |
| | $f_{nc}$ | -0.075 | 0.072 | 0.303 | 0.007 | 0.137 | 0.960 |
| | $f_{ac}$ | 0.418 | 0.175 | 0.017 | 0.294 | 0.131 | 0.025 |
| | $f_n$ | 0.342 | 0.048 | 0.000 | 0.343 | 0.052 | 0.000 |
| | $f_i$ | 0.126 | 0.030 | 0.000 | 0.115 | 0.036 | 0.001 |
| GPT-4o-2024-11-20 | $f_{sc}$ | 0.259 | 0.039 | 0.000 | 0.271 | 0.050 | 0.000 |
| | $f_{nc}$ | -0.065 | 0.082 | 0.432 | -0.050 | 0.121 | 0.680 |
| | $f_{ac}$ | 0.182 | 0.071 | 0.010 | 0.003 | 0.111 | 0.975 |
| | $f_n$ | 0.489 | 0.040 | 0.000 | 0.459 | 0.058 | 0.000 |
| | $f_i$ | 0.144 | 0.032 | 0.000 | 0.159 | 0.033 | 0.000 |
| GPT-4-0613 | $f_{sc}$ | 0.182 | 0.033 | 0.000 | 0.188 | 0.035 | 0.000 |
| | $f_{nc}$ | 0.237 | 0.088 | 0.007 | 0.264 | 0.124 | 0.034 |
| | $f_{ac}$ | -0.030 | 0.077 | 0.698 | -0.008 | 0.122 | 0.945 |
| | $f_n$ | 0.380 | 0.044 | 0.000 | 0.356 | 0.057 | 0.000 |
| | $f_i$ | 0.212 | 0.033 | 0.000 | 0.217 | 0.036 | 0.000 |

To further assess whether these contributions are truly causal, we conduct a *causal analysis*. We report the average treatment effect (ATE) and standard error (SE) of each factor to the overall accuracy from two estimation methods—ordinary least squares (OLS) [3] and propensity score matching (Matching) [44]—across different LLMs in Table 6. Overall, the findings are consistent with those of the factor analysis. However, one additional insight emerges: while $f_{ac}$ shows a consistently positive contribution in factor analysis, its causal effect on overall accuracy is not statistically significant for Qwen-32B, DeepSeek-V3, and again, the outlier GPT-4. This suggests that the predictive utility of $f_{ac}$ may be model-specific and conditional on the broader reasoning process.

### A.4.3   Case Study: Reasoning Steps under Different Settings

For the first axis of our case study, we present the reasoning steps of Claude under different settings—namely, vanilla, zero-shot CoT, manual CoT, and AC-REASON—in order to quantitatively assess the effectiveness of AC-REASON. The selected example is shown below. In this example, there are three main causal events: 1) $E_1$, Alex's miscommunication about the can color; 2) $E_2$, Alex uses A X200R; and 3) $E_3$, Benni unknowingly uses B Y33R following Tom's instruction. The outcome event $O$ is "the plants dry out". It is straightforward to identify that the conjunction of $E_2$ and $E_3$ constitutes a cause of $O$—that is, each is necessary but not sufficient and thus not a cause of $O$ on its own. A potential source of confusion arises from $E_3$: one might mistakenly judge it to be an individual cause of $O$ because it appears to violate the norm of Tom's instruction. However, this is incorrect. The relevant norm in this context is the instruction provided by Alex—that A X200R is in the green can and that Benni should use the fertilizer from the green can. Since Benni follows this instruction, $E_1$ does not involve a norm violation. Therefore, $E_1$ is merely part of a conjunctive

cause of $O$ and does not have the uniquely highest responsibility (by analyzing the temporal order), making the correct answer No (Line 21 of Algorithm 1).

---

**Example Sample**

**Story:** Tom has a huge garden and loves flowers. He employed two gardeners who take care of the plants on his 30 flower beds: Alex and Benni. Both can independently decide on their working hours and arrange who cares for which flower beds. Alex and Benni are very reliable and Tom is satisfied with their work. Nevertheless he wants to optimize the plant growth. Since Tom has read in a magazine that plants grow better when they are fertilized, he decides to let Alex and Benni fertilize his plants. The magazine recommends the use of the chemicals A X200R or B Y33R, since both are especially effective. However, Tom also read that it can damage plants when they are exposed to multiple different types of chemicals. Tom therefore decides that he only wants to use one fertilizer. He goes for A X200R. Tom instructs Alex and Benni to buy the chemical A X200R and to use only this fertilizer. Alex volunteers for buying several bottles of this chemical for Benni and himself. After a few weeks, Tom goes for a walk in his garden. He realizes that some of his plants are much prettier and bigger than before. However, he also realizes that some of his plants have lost their beautiful color and are dried up. That makes Tom very sad and reflective. He wonders whether the drying of his plants might have something to do with the fertilization. He wants to investigate this matter and talks to Alex and Benni. Alex tells him that he followed Tom's instruction: "I only bought and used the chemical A X200R which I had funneled into the blue can." Benni suddenly is startled and says to Alex: "What? You funneled A X200R into the blue can? But you told me you had funneled it into the green can! That's why I always used the green can!" Alex replies: "Did I? Then I am sorry!" Tom remembers that he had filled B Y33R in a green can - long before he had read about the chemicals in his magazine. He had never used it. So Benni must have accidentally, without knowing it, applied the chemical B Y33R, whereas only Alex applied A X200R. Tom realizes that the plants dried up in the flower beds on which both A X200R and B Y33R were applied by the gardeners.
**Query:** Did Benni cause the plant to dry out?
**Gold Answer:** No.

---

As shown below, the vanilla, zero-shot CoT, and manual CoT settings exhibit similar patterns. On the one hand, Claude demonstrates the ability to identify causal and outcome events without explicit prompting and can also take into account factors such as intention. This indicates that the first two steps of AC-REASON are implicit capabilities of LLMs. However, Claude falls short in two key areas under these settings: 1) it fails to comprehensively consider all relevant factors—e.g., sufficiency and normality are entirely overlooked despite their importance in this example; and 2) it lacks a structured mechanism to synthesize its causal analysis into a coherent decision. Consequently, Claude disproportionately relies on naive counterfactual dependence (i.e., $f_{nc}$), which undermines its reasoning quality. We have also seen similar patterns from other models, such as DeepSeek-R1. In contrast, under the AC-REASON setting, Claude performs all key reasoning steps correctly. It first identifies the focal causal event, "Benni uses green can", then accurately infers the factor values for both conjunctive causal events: "Benni uses green can" and "Alex uses blue can". Following this, it correctly traverses Algorithm 1 to reach the final decision. Specifically, "Benni uses green can" satisfies $\neg f_{sc} \wedge f_{ac}$ (Line 15), does not satisfy $f_n \vee f_i$ (Line 16), and is not assigned the uniquely highest responsibility due to occurring later than other causal events (Lines 20-21). Thus, it leads to the correct answer. Although Claude identifies three additional causal events under this setting, they are inconsequential, as all 5 causal factors take on False. This indicates that these events can be safely removed from the causal setting without affecting the outcome. The comparison between AC-REASON and the other three settings is further supported by the findings in Appendix A.4.2: when equipped with AC-REASON, Claude correctly utilizes all factors except $f_{nc}$ and exhibits faithful reasoning, whereas in all other settings, its reasoning heavily depends on $f_{nc}$.

---

**Claude Vanilla**

To answer this question, we need to carefully analyze the causal chain of events and consider the role of Benni's actions in the outcome.

---

Let's break down the key points:

1. Tom decided to use only one fertilizer (A X200R) to avoid damaging plants with multiple chemicals.
2. Alex bought A X200R and funneled it into the blue can, but mistakenly told Benni it was in the green can.
3. Benni, believing Alex's misinformation, used the fertilizer from the green can.
4. The green can actually contained B Y33R, which Tom had filled long ago and never used.
5. The plants dried up in flower beds where both A X200R and B Y33R were applied.

Benni's action of using the fertilizer from the green can (unknowingly B Y33R) was a crucial part of the causal chain that led to the plants drying out. If Benni had not applied B Y33R, the plants would not have been exposed to both chemicals, which was the condition that caused the drying.

However, it's important to note that Benni's action was based on misinformation from Alex and was unintentional. Benni believed he was following Tom's instructions and using the correct fertilizer.

Despite the unintentional nature of Benni's action, from a strict causal perspective, Benni's application of B Y33R was a necessary condition for the harmful outcome.

Therefore, the answer is:

Yes

Benni did cause the plants to dry out, albeit unintentionally and as part of a chain of events involving misinformation and misunderstanding.

673

---

### Claude + zero-shot CoT

To answer this question, let's break down the key events and analyze the causal chain:
1. Tom decided to use only one fertilizer (A X200R) to avoid damaging plants with multiple chemicals. 2. Alex bought A X200R and funneled it into a blue can. 3. Alex mistakenly told Benni that he had put A X200R in the green can. 4. Benni, trusting Alex's information, used the fertilizer from the green can. 5. The green can actually contained B Y33R, which Tom had filled long ago and never used. 6. The plants dried up in flower beds where both A X200R and B Y33R were applied.

Now, let's consider Benni's role in this causal chain:

1. Benni did not intentionally use B Y33R; he thought he was using A X200R based on Alex's misinformation.
2. Benni's action of using the fertilizer from the green can was a necessary component in the causal chain that led to the plants drying out.
3. If Benni had not used the fertilizer from the green can (B Y33R), the plants would not have been exposed to both chemicals and would not have dried out.
4. While Benni's action was based on misinformation, he still played a crucial role in the outcome.
5. In causal judgment, we often consider whether an action was a necessary part of the causal chain, regardless of the actor's intentions or knowledge.

Given these considerations, we can conclude that Benni did cause the plants to dry out, even though it was unintentional and based on misinformation.

Answer: Yes

Benni's actions were a necessary part of the causal chain that led to the plants drying out, even though he was not aware of the true nature of the fertilizer he was using. In causal analysis, the unintended consequences of an action can still be attributed to the actor if their action was necessary for the outcome to occur.

674

675

**Claude + manual CoT**

To answer this question, let's analyze the key points from the story:

1. Tom instructed both gardeners to use only A X200R fertilizer.
2. Alex bought A X200R and put it in the blue can.
3. Benni mistakenly thought A X200R was in the green can, which actually contained B Y33R.
4. The plants dried up in flower beds where both A X200R and B Y33R were applied.
5. Benni unknowingly used B Y33R from the green can.

Given these facts, we can conclude that Benni did indeed cause the plants to dry out, albeit unintentionally. Here's why:

1. Benni's use of B Y33R, combined with Alex's use of A X200R, created the condition that led to the plants drying out.
2. If Benni had not used the fertilizer from the green can (B Y33R), the plants would not have been exposed to both chemicals, which was the cause of the damage.
3. Even though Benni's actions were accidental and based on misinformation, they still contributed directly to the outcome.

Therefore, the answer is: Yes
Benni caused the plants to dry out, even though it was unintentional and based on a misunderstanding. In causal reasoning, the intent is not always necessary for establishing causation; the action and its consequences are what matter most.

```
{
    "causal_events": {
        "Tom instructs to use A X200R": {
            "occur": True,
            "order": 0,
            "focal": False,
            "sufficient": False,
            "necessary": False,
            "halpern_pearl": False,
            "norm_violated": False,
            "behavior_intended": False
        },
        "Alex buys A X200R": {
            "occur": True,
            "order": 1,
            "focal": False,
            "sufficient": False,
            "necessary": False,
            "halpern_pearl": False,
            "norm_violated": False,
            "behavior_intended": False
        },
        "Alex funnels A X200R into blue can": {
            "occur": True,
            "order": 2,
            "focal": False,
            "sufficient": False,
            "necessary": False,
            "halpern_pearl": False,
            "norm_violated": False,
```

676

```
                "behavior_intended": False
            },
            "Alex tells Benni about green can": {
                "occur": True,
                "order": 3,
                "focal": False,
                "sufficient": False,
                "necessary": True,
                "halpern_pearl": True,
                "norm_violated": False,
                "behavior_intended": False
            },
            "Benni uses green can": {
                "occur": True,
                "order": 4,
                "focal": True,
                "sufficient": False,
                "necessary": True,
                "halpern_pearl": True,
                "norm_violated": False,
                "behavior_intended": False
            },
            "Alex uses blue can": {
                "occur": True,
                "order": 4,
                "focal": False,
                "sufficient": False,
                "necessary": True,
                "halpern_pearl": True,
                "norm_violated": False,
                "behavior_intended": False
            }
        },
        "outcome_event": {
            "Plants dry out": {
                "occur": True,
                "order": 5
            }
        }
    }
}
```

Explanation:
No. Based on the story, "Benni uses green can" is not more responsible relative to the other events in the list. All three events ("Benni uses green can", "Alex tells Benni about green can", and "Alex uses blue can") occur in sequence with "Alex tells Benni about green can" happening first (order 3), followed by both "Benni uses green can" and "Alex uses blue can" occurring simultaneously (both order 4). Since "Alex tells Benni about green can" occurs earlier, it would have more responsibility according to the temporal order criterion. Between "Benni uses green can" and "Alex uses blue can", they have equal responsibility since they occur simultaneously (same order 4).

### A.4.4  Case Study: Seed Samples vs. Generated Samples

As for the second axis, we present examples comparing seed samples with generated samples to demonstrate that AC-BENCH is both more challenging and diverse. The increased complexity and diversity primarily stem from the second step of our generation pipeline (see Appendix A.7): 1) Through addition, new details with spurious correlations may be introduced, potentially distracting models from the actual causal structure. For instance, in Example 1, the story of the seed sample

contains only events relevant to the causal setting. In contrast, the generated sample includes an additional, irrelevant event—"Sophia prepares the main course and appetizers". This extraneous detail may mislead models into inferring a spurious correlation between this event and the outcome, thereby increasing the reasoning difficulty. 2) Through removal, important but non-essential causal cues—such as explicit conjunctive statements like "when $E_1$ and $E_2$ occur, $O$ will occur"—may be omitted. For instance, in Example 2, the seed sample explicitly states that "the machine will short circuit if both the black wire and the red wire touch the battery at the same time". In contrast, the corresponding generated sample omits this explicit conjunctive specification, instead conveying it only implicitly. Such omissions increase the reasoning complexity of the sample. 3) Through reorganization, the structure of the story may become more diverse. In both Example 1 and 2, the seed sample and its corresponding generated sample differ in multiple aspects, including the story setting, the addition or removal of specific details, and the organization of individual sentences as well as the overall paragraph structure. As a result, the generated samples in AC-BENCH pose greater challenges and exhibit higher diversity than their seed counterparts. This increased difficulty is also reflected in model performance: across all models, accuracy on causation queries declines in AC-BENCH compared with in BBH-CJ, and the gains attributed to AC-REASON are reduced accordingly.

---

**Example 1: Addition leads to spurious correlations.**

### Seed Sample
**Story:** Louie is playing a game of basketball, and he made a bet with his friends who are watching on the sidelines. If Louie either makes a layup or makes a 3-point shot during the game, then he'll win $100. Just when the game started, Louie immediately got the ball at the 3-point line. He looked to the basket, dribbled in, and then made a layup right at the beginning of the game. Louie and his friends continued playing, but as hard as he tried, Louie couldn't make another shot. And then right at the end of the game as the clock was winding down, Louie got the ball at the 3-point line. He looked to the basket, focused his shot, and made a 3-point shot right at the buzzer. Then the game ended. Because Louie would win $100 if he either made a layup or a 3-point shot, Louie won $100.
**Query:** Did Louie win the $100 bet because he made the layup?
**Reasoning:**

```
{
    "causal_events": {
        "Louie makes a layup": {
            "occur": true,
            "order": 0,
            "focal": true,
            "sufficient": true,
            "necessary": false,
            "halpern_pearl": false,
            "norm_violated": false,
            "behavior_intended": true
        },
        "Louie makes a 3-point shot": {
            "occur": true,
            "order": 1,
            "focal": true,
            "sufficient": true,
            "necessary": false,
            "halpern_pearl": false,
            "norm_violated": false,
            "behavior_intended": true
        }
    },
    "outcome_event": {
        "Louie wins \$100": {
            "occur": true,
```

```
            "order": 2
        }
    }
}
```
**Answer:** Yes

---

### Generated Sample
**Story:** Sophia has invited friends over for a cozy dinner evening at her apartment, where she plans to showcase her cooking skills. The highlight of the night, according to her invitation, is the dessert, which she promised to prepare herself. The party will feature dessert if Sophia either bakes a cake or bakes cookies. Sophia decided to begin the day by focusing on the most intricate dessert: the cake. She spent hours meticulously measuring ingredients, frosting layers, and decorating the top with a swirl of fresh whipped cream. It turned out beautifully, and she proudly displayed it on the dining table. Feeling confident, Sophia dedicated her remaining time to preparing the main course and appetizers. As the guests began sending texts that they were on their way, Sophia realized she had just enough time to bake a quick batch of cookies. They were ready just as her first guests entered the apartment. With both options available, the party was a success, and dessert was served. Since the party would have dessert if Sophia either baked cookies or a cake, there was no shortage of delicious treats to enjoy.
**Query:** Did the dinner party have dessert because Sophia baked cookies?
**Reasoning:**
```
{
    "causal_events": {
        "Sophia bakes cookies": {
            "occur": true,
            "order": 1,
            "focal": true,
            "sufficient": true,
            "necessary": false,
            "halpern_pearl": false,
            "norm_violated": false,
            "behavior_intended": true
        },
        "Sophia bakes a cake": {
            "occur": true,
            "order": 0,
            "focal": true,
            "sufficient": true,
            "necessary": false,
            "halpern_pearl": false,
            "norm_violated": false,
            "behavior_intended": true
        }
    },
    "outcome_event": {
        "The party has dessert": {
            "occur": true,
            "order": 2
        }
    }
}
```
**Answer:** No

702

### Seed Sample

**Story:** A machine is set up in such a way that it will short circuit if both the black wire and the red wire touch the battery at the same time. The machine will not short circuit if just one of these wires touches the battery. The black wire is designated as the one that is supposed to touch the battery, while the red wire is supposed to remain in some other part of the machine. One day, the black wire and the red wire both end up touching the battery at the same time. There is a short circuit.

**Query:** Did the black wire cause the short circuit?

**Reasoning:**

```
{
    "causal_events": {
        "The black wire touches the battery": {
            "occur": true,
            "order": 0,
            "focal": true,
            "sufficient": false,
            "necessary": true,
            "halpern_pearl": true,
            "norm_violated": false,
            "behavior_intended": false
        },
        "The red wire touches the battery": {
            "occur": true,
            "order": 0,
            "focal": false,
            "sufficient": false,
            "necessary": true,
            "halpern_pearl": true,
            "norm_violated": true,
            "behavior_intended": false
        }
    },
    "outcome_event": {
        "The machine short circuits": {
            "occur": true,
            "order": 1
        }
    }
}
```

**Answer:** No

---

### Generated Sample

**Story:** In a water management system for a large greenhouse, there are two valves controlling water supply to a central storage tank. Valve A is part of the main operational circuit, intended to manage daily water flow. Valve B, on the other hand, serves as an emergency bypass valve that should remain shut to prevent mixing excess water into the system. Under normal circumstances, if just one valve is open, the tank functions properly without incident. One morning, during a routine inspection, both Valve A and Valve B are mistakenly left open for hours, allowing water from two separate sources to flow into the tank concurrently. As a result, the tank exceeds its capacity and floods the greenhouse floor, causing damage.

**Query:** Did Valve A cause the overflow?

**Reasoning:**

```
{
    "causal_events": {
```

703

```
                "Valve A is open": {
                    "occur": true,
                    "order": 0,
                    "focal": true,
                    "sufficient": false,
                    "necessary": true,
                    "halpern_pearl": true,
                    "norm_violated": false,
                    "behavior_intended": false
                },
                "Valve B is open": {
                    "occur": true,
                    "order": 0,
                    "focal": false,
                    "sufficient": false,
                    "necessary": true,
                    "halpern_pearl": true,
                    "norm_violated": true,
                    "behavior_intended": false
                }
            },
            "outcome_event": {
                "The storage tank overflows": {
                    "occur": true,
                    "order": 1
                }
            }
        }
```

**Answer:** No

## A.5   Details of AC-REASON

### A.5.1   Generation of Explanations

In Algorithm 1, an explanation is generated for each answer. Most explanations are produced using predefined templates (Lines 4, 5, 9-10 and 16). For responsibility-related decisions (Lines 13-14 and 20-21), the explanations are generated by the LLM responsible for determining responsibility. First, we present the templates for producing explanations. In each template, the placeholder $E$ is replaced with the actual causal event in the current for-loop, and $O$ is replaced with the actual outcome event.

Ln. 04. $E$ is a cause of $O$, since $E$ is both sufficient and necessary.

Ln. 05. $E$ is not a cause of $O$, since $E$ is neither sufficient nor necessary.

Ln. 09. $E$ is a cause of $O$, since $E$ occurs the earliest in a disjunctive causal structure.

Ln. 10. $E$ is not a cause of $O$, since $E$ does not occur the earliest in a disjunctive causal structure.

Ln. 16. $E$ is a cause of $O$, since $E$ is an actual cause and it violates a norm or is an intended behavior of an agent.

Second, we provide examples of LLM-generated explanations.

> **Example for Line 13.**
>
> Yes, the focal causal event "Billy arrives at 9 am" is equally responsible relative to the other causal event "Suzy arrives at 9 am". Both Billy and Suzy arrived at the same time, triggering the motion detector. Although Billy was explicitly told not to come, Suzy was also explicitly told to come at exactly 9 am, making both of them equally responsible for the outcome.

No. In the story, Drew, Kylie, and Oliver all ordered coffee on the same day, and any one of them ordering coffee would have been sufficient for the coffee shop to make a profit. However, Drew's action of ordering coffee on a Tuesday was unexpected and violated the norm, as he usually doesn't order coffee on Tuesdays. This makes Drew's action more noticeable, but it doesn't make him more responsible for the coffee shop's profit than Kylie or Oliver, who also ordered coffee.

Yes, the focal causal event "Brother steps on a slippery patch" is more responsible relative to other causal events in the list. This is because it occurs later in the sequence of events, closer to the outcome event. The brother stepping on the slippery patch directly leads to him knocking over the lamp, which then fractures the water pipe, causing the damage. The other events, while they set the stage for this to happen, are further removed from the outcome event in terms of temporal order.

No. Based on the story, Alice's action of logging in at 9.30 am is not more responsible than Zoe's action of logging in at 9 am for the outcome event. This is because both actions are necessary for the outcome to occur, and Zoe's action, which violates the company policy, happens first. Therefore, Zoe's action has a higher temporal order and is considered more responsible for the outcome.

## A.6    Details of AC-BENCH

### A.6.1    Details of Data Cleaning

We perform the following operations to clean the BBH-CJ dataset:

Op. 1.  Remove *duplicate* samples.

Op. 2.  Remove *erroneous* samples, which are considered invalid—for example, those missing critical background information.

Op. 3.  Remove *irrelevant* samples. In BBH-CJ, there are two types of queries: 1) **causation**, e.g., *Did John's job cause his premature death?* 2) **intention**, e.g. *Did the CEO intentionally harm the environment?* Specifically, we exclude samples querying about intention, since our focus is on actual causal relationships between real-world events.

Op. 4.  Correct *partially flawed* samples. These are samples with issues in the story background, query, or answer. In many cases, errors in the query and answer co-occur—for instance, a duplicated or malformed query often results in an incorrect answer.

The goal of this data cleaning process is to reduce noise in BBH-CJ and retain only high-quality samples relevant to AC. We also provide examples illustrating the last three type of operation below.

**Erroneous Sample.**

The "motion detector" mentioned in the question has not appeared in the story background.
**Story:** Suzy and Billy are working on a project that is very important for our nation's security. The boss tells Suzy: "Be sure that you are here at exactly 9 am. It is absolutely essential that you arrive at that time." Then he tells Billy: "Be sure that you do not come in at all tomorrow morning. It is absolutely essential that you not appear at that time." Both Billy and Suzy arrive at 9 am.
**Query:** Did Billy cause the motion detector to go off?
**Answer:** No

In the third sentence of the story, "work emails" is mistakenly written as "spam emails".
**Story:** Billy and Suzy work for the same company. They work in different rooms, and both of them sometimes need to access the central computer of the company. Nobody at the company is aware that if two people are logged into the central computer at the same time, some spam emails containing important customer information are immediately deleted from the central computer. In order to make sure that two people are available to answer phone calls during designated calling hours, the company issued the following official policy: Billy and Suzy are both permitted to log into the central computer in the mornings, and neither of them are permitted to log into the central computer in the afternoons. Today at 9 am, Billy and Suzy both log into the central computer at the same time. Immediately, some work emails containing important customer information are deleted from the central computer.
**Query:** Did Suzy cause the central computer to delete some work emails containing important customer information?
**Answer:** Yes

### A.6.2 Details of Data Annotation

The labeling process does not involve crowdsourcing; instead, it is conducted by a master's student specializing in actual causality with LLMs. The annotation criteria are presented in Table 7, which are primarily derived from established factors in the causal judgment literature [40] and formal definitions from the actual causality domain [16]. The annotation process follows the pipeline below:

Step 1. Annotate $\mathcal{C}$ and $O$. We first manually annotate the events. Then, we use GPT-4 to perform event detection and refine our annotations by comparing them with GPT-4's results.

Step 2. Annotate $(f_{occur}, f_{order})$ based on the annotation guideline. The occurrences and temporal orders of events are usually explicit in the story, making them relatively straightforward to label.

Step 3. Annotate $(f_{sc}, f_{nc}, f_{ac}, f_n, f_i)$ based on the annotation guideline. This step is more labor-intensive due to the complexity of the factors (especially $f_{ac}$). First, we group the factors into two sets: $(f_{sc}, f_{nc}, f_{ac})$ and $(f_n, f_i)$. In the first group, the annotation follows the order $f_{nc} \rightarrow f_{ac} \rightarrow f_{sc}$. This order is based on dependency: $f_{nc}$ is the simplest to determine, and if it holds, $f_{ac}$ necessarily holds as well; in turn $f_{sc}$ depends on $f_{ac}$. Only when $\neg f_{nc}$ holds do we need to assess $f_{ac}$ more carefully by considering *contingencies* in AC2. In AC2, contingencies refer to holding the values of $W$ fixed at their original values when intervening to set $X$ to $x'$. The second group, $(f_n, f_i)$, is relatively straightforward to annotate, as these factors are also explicit in the story. Second, to further alleviate cognitive load, we input the annotated events into GPT-4 and use its predictions as auxiliary references. We then carefully revise the annotations in accordance with the definitions in the annotation guideline.

For each step, we manually verify the annotations multiple times to ensure high-quality labeling.

### A.6.3 Details of Data Verification

The high quality of AC-BENCH is ensured by the following factors:

Table 7: Annotation guideline.

| Factors | Definitions |
|---|---|
| **Occurrence** ($f_{occur}^E$) | |
| *True* | $E$ actually occurs in the causal setting. |
| *False* | Otherwise. |
| **Temporal Order** ($f_{order}^E$) | |
| / | An integer starting from 0, denoting the temporal order of $E$ relative to other events in the causal setting. If two events occur simultaneously, they should share the same $f_{order}$. |
| **Sufficient Causality** ($f_{sc}^E$) | |
| *True* | $E$ is a sufficient cause of $O$ by Definition 2. If 1) $f_{ac}^E$ holds and 2) $E$ always suffices to cause $O$ while changing the $f_{occur}$ factor of other causal events, then $f_{sc}^E$ holds. [Proof: SC1 holds if $f_{occur}^E \wedge f_{occur}^O$ holds; SC2 holds if $f_{ac}^E$ holds; SC3 holds by counterfactual reasoning on causal events other than $E$; SC4 already holds since we only annotate minimal/atomic and non-overlapping causal events. Also, if $f_{ac}^E$ holds, $f_{occur}^E \wedge f_{occur}^O$ holds.] |
| *False* | Otherwise. |
| **Necessary Causality** ($f_{nc}^E$) | |
| *True* | $E$ is a necessary cause of $O$ by the but-for definition. If but for $E$, $O$ would not have occurred, then $f_{nc}^E$ holds. |
| *False* | Otherwise. |
| **Halpern-Pearl Causality** ($f_{ac}^E$) | |
| *True* | $E$ is an actual cause of $O$ by Definition 1. If 1) $f_{occur}^E \wedge f_{occur}^O$ holds and 2) $f_{nc}^E$ holds while allowing contingencies based on AC2, then $f_{ac}^E$ holds. [Proof: AC1 holds if $f_{occur}^E \wedge f_{occur}^O$ holds; AC2 holds if $f_{nc}^E$ holds, while allowing contingencies; AC3 already holds since we only annotate minimal/atomic and non-overlapping causal events.] |
| *False* | Otherwise. |
| **Normality** ($f_n^E$) | |
| *True* | $E$ violates a prescriptive or descriptive norm. |
| *False* | Otherwise. |
| **Intention** ($f_i^E$) | |
| *True* | $E$ is an agent's behavior that is intentional, and $O$ is adverse and foreseeable. |
| *False* | Otherwise. |

**Reliable data source.** AC-BENCH is built upon BBH-CJ [50], which contains 187 challenging causal judgment queries selected from Big-Bench Causal Judgment (BB-CJ) [49]. As indicated in its Github repository, the stories, queries, and answers in BB-CJ are curated from 30 papers published between 1989 and 2021. Each paper involves rigorous human experiments, and the ground-truth binary answers are derived from these experimental results. Therefore, the samples in BBH-CJ can be considered fundamentally reliable.

**Thorough data cleaning.** We perform extensive data cleaning on BBH-CJ, as detailed in Appendix A.6. This includes removing duplicate, erroneous, and irrelevant samples, as well as correcting partially flawed samples, thereby significantly reducing noise in the dataset.

**Step-by-step, LLM-assisted labeling with multi-round manual verification.** While the average human accuracy on BBH-CJ is only 69.60% as reported in Big-Bench Hard [50], this does not reflect the quality of our annotations. First, we decompose each problem into intermediate reasoning steps, allowing annotators to better understand and process each sample. Second, we leverage LLM-generated reasoning steps as auxiliary information to reduce annotator cognitive load, as described in Appendix A.6. Finally, after labeling, each annotation is manually reviewed multiple times to ensure the quality of the resulting "seed samples" used for data generation.

**Post-generation quality control via automatic and manual checks.** For *automated validation*, we implement code to automatically check whether the reasoning logic, i.e., causal setting and causal factors, of each generated sample is consistent with its seed sample. Fewer than 1% of samples are found inconsistent (mostly involving incorrect values for $f_i$), which are then manually corrected. Additionally, we verify whether the causal factors in each generated sample corresponds to a correct gold answer using Algorithm 1; all samples pass this automatic check. For *manual inspection*, we randomly sample ~33.3% (310 out of 930) generated samples and evaluate whether 1) the story contains sufficient information for actual causal reasoning, 2) the reasoning logic is consistent with

the story, and 3) the gold answer is correct. Only 2 samples (0.6%) are found to lack essential reasoning information; we regenerate and verify these samples.

## A.7 Prompts

### A.7.1 Prompts for the Baselines

---

**Vanilla**

[SYSTEM] You are an expert in the field of actual causality and causal judgment. Given the story and query of a logic-based causal judgment problem, you can effectively solve it.
**Story**: {story}
**Query**: {query}
Answer (Yes or No?):

---

**Zero-shot CoT**

[SYSTEM] You are an expert in the field of actual causality and causal judgment. Given the story and query of a logic-based causal judgment problem, you can effectively solve it.
**Story**: {story}
**Query**: {query}
Let's think step by step.
Answer (Yes or No?):

---

**Manual CoT**

[SYSTEM] You are an expert in the field of actual causality and causal judgment. Given the story and query of a logic-based causal judgment problem, you can effectively solve it.
Here we will provide three chain-of-thought examplars, followed by a binary question that needs to be answered.
**Story**: {story1}
**Query**: {query1}
**Answer (with chain of thought)**: {answer1}
**Story**: {story2}
**Query**: {query2}
**Answer (with chain of thought)**: {answer2}
**Story**: {story3}
**Query**: {query3}
**Answer (with chain of thought)**: {answer3}
**Story**: {story}
**Query**: {query}
Answer (Yes or No?):

---

### A.7.2 Prompts for AC-REASON

---

**Causal Setting Establishment**

[SYSTEM] You are an expert in the field of actual causality and causal judgment. Given the story and query of a logic-based causal judgment problem, you can effectively assist in solving the problem following the instructions provided.
**Story**: {story}
**Query**: {query}
Based on the story and query of a logic-based causal judgment problem, establish the causal setting as follows.

    1. Summarize the **causal events** and **outcome event** based on the story and query.

       – The causal events should causally contribute to the outcome event.

---

- The causal events should be minimal/atomic and non-overlapping.

2. Label the **occurrences** and **temporal orders** of events based on the story.
   - Label `occur` as `true` if an event actually occurs.
   - Label `order` as an integer starting from 0, where simultaneous events should share the same `order`.

3. Label the **focal causal events** based on the query.
   - If the query asks whether a causal event causes the outcome event, the causal event has `focal = true`.
   - If the query asks whether an agent causes the outcome event, all causal events reflecting the agent's behaviors have `focal = true`.

Return the causal setting in the following JSON format:

```
{
    "causal_events": {
        CAUSAL_EVENT: {
            "occur": True/False,
            "order": ORDER,
            "focal": True/False
        },
        ...
    },
    "outcome_event": {
        OUTCOME_EVENT: {
            "occur": True/False,
            "order": ORDER
        }
    }
}
```

Return only the JSON, without any extra information.

799

## Causal Factors Analysis

Based on the story and causal setting, reason about the values of the following causal factors for each causal event.

1. `sufficient = true` if in the story, had a causal event occurred, the outcome event would have occurred, even if other causal events had occurred differently.

2. `necessary = true` if in the story, had a causal event not occurred, the outcome event would not have occurred.

3. `halpern_pearl = true` if in the story, had a causal event not occurred, the outcome event would not have occurred, while allowing at least a subset of events in the causal setting to remain occurred had a causal event not occurred.

   * `sufficient = true`, `necessary = true`, and `halpern_pearl = true` can be satisfied through a path from a causal event to the outcome event, passing through other causal events.

4. `norm_violated = true` if in the story, a causal event violates norms, such as statistical modes, moral codes, laws, policies, or proper functioning in machines or organisms.

5. `behavior_intended = true` if in the story, a causal event is an agent's behavior and the agent is aware of the potential consequences of their action and knowingly performs an action that leads to a foreseeable bad outcome.

Return the values of factors for causal events in the following JSON format:

```
{
```

800

```
    CAUSAL_EVENT: {
        "sufficient": True/False,
        "necessary": True/False,
        "halpern_pearl": True/False,
        "norm_violated": True/False,
        "behavior_intended": True/False
    },
    ...
}
```
Return only the JSON, without any extra information.

---

### Determining $f_{resp}$ (Line 12)

Define responsibility as the relative degree (more, less, or equally) to which a causal event causally contributes to the outcome event, relative to other causal events specified. Here, assume responsibility is only determined by normality (`norm_violated`) and intention (`behavior_intended`).
Return Yes if based on the story, the focal causal event "{focal_event}" is equally or more responsible relative to other causal events in the list {S_list}, else No. Then, explain briefly based on the story.

---

### Determining $f_{resp}$ (Line 19)

Define responsibility as the relative degree (more, less, or equally) to which a causal event causally contributes to the outcome event, relative to other causal events specified. Here, assume responsibility is only determined by temporal order (`order`).
Return Yes if based on the story, the focal causal event "{focal_event}" is more responsible relative to other causal events in the list {H_list}, else No. Then, explain briefly based on the story.

## A.7.3 Prompts for AC-BENCH

### Data Generation (Stage 1)

**Data Example**: {data_example}
In the provided data sample: `story` is the story background of a logic-based causal judgment problem; `reasoning` details the reasoning process, including causal/outcome events and their factor values; `queries` are causality-related queries to the problem, each links to one or more focal causal events in `reasoning`; `answers` are Yes/No answers to the queries.
Generate a new data example as follows:

1. `story`: Rewrite with a different real-world setting while preserving the reasoning logic, i.e., only rephrase causal/outcome event descriptions.

2. `queries`: Formulate queries for the new story and identify the corresponding focal causal events.

3. `reasoning`: Generate causal/outcome events and their factors for the new story while keeping factor values unchanged.

4. `answers`: Provide Yes/No answers to the queries.

Return the new data example in the same JSON format as the original one.
Return only the JSON, without any extra information.

> **Data Generation (Stage 2)**
>
> Based on the generated data example, refine the new story to make it more distinctive while keeping `reasoning` unchanged:
>
> 1. Increase Details: Add relevant details to enhance the new story setting.
>
> 2. Remove Details: Eliminate elements that are only relevant to the original story.
>
> 3. Reorganize Structure: Adjust the narrative flow to avoid mirroring the original structure.
>
> Return the modified data example in the same JSON format as the original one.
> Return only the JSON, without any extra information.

## A.8 Limitations and Future Work

First, the AC-REASON framework remains incomplete. It incorporates only several commonly studied factors in actual causality and causal judgment, potentially overlooking more fine-grained or domain-specific factors that may influence reasoning. Moreover, it is currently limited to processing only one minimal causal event at a time with our Algorithm 1 and lacks the capability to account for conjunctive causal events as candidate causes. Second, the BBH-CJ dataset comprises samples drawn from research papers published between 1989 and 2021. We have not yet included samples from newly emerged papers. As for future work, we plan to do the following. First, Algorithm 1 can be enhanced by incorporating additional factors and refining its reasoning rules. Second, we will try to apply our framework in domain-specific scenarios, such as legal reasoning, where actual causality and causal judgment play a central role. Finally, we plan to expand our benchmark in response to newly emerged research works in these fields.

## A.9 Broader Impacts

The proposed AC-REASON framework introduces a theory-grounded approach to actual causality (AC) reasoning with LLMs, which has both promising societal benefits and potential risks. On the positive side, AC-REASON enhances the interpretability and formal rigor of LLM-based causal reasoning (CR). This advancement can support responsible decision-making in high-stakes domains such as legal analysis, policy evaluation, and scientific explanation—areas where precise attribution of causality is essential. Furthermore, the introduction of AC-BENCH provides the community with a more challenging and structured benchmark, promoting research into deeper, more human-aligned CR. However, the ability to assign blame or responsibility with greater formal precision also raises ethical concerns. In legal or social contexts, misuse of such models—especially without adequate human oversight—could lead to unjustified attributions of responsibility or reinforce biased norms embedded in training data. Moreover, the model's interpretability may lend unwarranted credibility to flawed outputs if users fail to critically evaluate them. To mitigate these risks, future deployments should incorporate transparency about model limitations, ensure human-in-the-loop oversight in critical applications, and explore fairness-aware variants of causal factor modeling. Additionally, controlled access to AC-REASON's reasoning capabilities may help prevent misuse in manipulative or adversarial settings.

