# OpenReview forum: "AC-Reason: Towards Theory-Guided Actual Causality Reasoning with Large Language Models"
_NeurIPS.cc/2025/Conference — Submitted to NeurIPS 2025_

### Official Review · Reviewer_ojMy · 2025-06-30

**Clarity:** 2
**Significance:** 2
**Originality:** 3
**Rating:** 4
**Confidence:** 3

**Summary:**

The authors claim three contributions:
1. The authors introduce the AC-REASON framework, which is claimed to combine LLMs with AC theory for the purpose of reasoning about actual causation.
2. The authors construct the AC-BENCH benchmark, which focuses solely on actual causal relationships. It scales BBH-CJ to ~1K samples and incorporates more challenging samples, enabling more comprehensive evaluation.
3. Experiments show that AC-REASON consistently improves performance across various LLMs, outperforming other baselines. Also, the integration of AC theory into LLMs via our algorithm is proved to be highly effective by the ablation study.

**Questions:**

1. What measurable / reproducible improvements can be shown when comparing AC-BENCH to BBH-CJ? Can precise details on the outcomes (e.g. # samples removed, # samples modified) of the operations in A.6.1 be described?
2. Human performance is 69.60% on BBH-CJ. What about on AC-BENCH?
3. Why does LLM performance go down from BBH-CJ to AC-BENCH? Can we be confident that human performance is still improved upon in AC-BENCH?
4. Is the improvement in accuracy statistically significant when using an approximate randomization test? What is the p-value? Is the p-value still significant when adjusting for the number of experimental trials (e.g., using Bonferroni correction)? I ask because there are fewer than 200 samples in BBH-CJ.
5. How do LLMs perform on the original subset of AC-BENCH that is not synthetically generated with data augmentation - does it differ from performance on the generated subset?
6. Does BBH-CJ contain questions about multivariate-causation - conjunctive or disjunctive settings? Were these filtered out, or were they not included in the original benchmark?

**Ethical Concerns:**

["NO or VERY MINOR ethics concerns only"]

**Final Justification:**

1. AC-REASON improves performance on an existing benchmark with statistical significance (as shown in the rebuttal).
2. The quality of AC-BENCH is suspect because 1) human performance is consistently low at 66%, 2) the literature often disagrees in judgments of actual causation, and 3) labeling was not crowd-sourced and was performed by a single human - inter-annotator reliability measures cannot be calculated.

**Limitations:**

Yes; the authors acknowledge the method is limited to a single minimal causal event at a time.

**Paper Formatting Concerns:**

None.

**Quality:**

3

**Strengths And Weaknesses:**

Quality: The authors show a performance improvement on BBH-CJ. The reliability of the samples in AC-BENCH seems questionable because "The labeling process does not involve crowdsourcing; instead, it is conducted by a master’s student specializing in actual causality with LLMs." For this reason, it is unclear what correctness guarantees the dataset has, particularly given that disagreement in opinions on actual causation is common [1].

Clarity: The contributions, methods, and results are stated clearly.

Significance: The work empirically improves upon a previous benchmark and claims to cleans this dataset.

Originality: The concept of an LLM-based AC reasoning is novel to my knowledge.

[1] Icard, Thomas F., Jonathan F. Kominsky, and Joshua Knobe. "Normality and actual causal strength." Cognition 161 (2017): 80-93.

---

> ### Author Rebuttal · Authors · 2025-07-31
>
> 1. **Reliability of AC-Bench.** AC-Bench is built upon the Causal Judgment subset of Big-Bench Hard [1], which itself originates from Big-Bench [2]. Big-Bench involves collecting 190 stories from 30 papers published between 1989 and 2021, each of which conducted rigorous human experiments to establish ground-truth causal judgments [2]. While minor errors may arise during data collection, such issues are addressed in our data cleaning process (Appendix A.6.1). Therefore, we regard the stories and answers in the Causal Judgment subset of both Big-Bench and Big-Bench Hard as reliable, with no fundamental disagreements about "actual causation" or "causal judgment". As for the labeling of reasoning steps, Big-Bench also includes a "comment" field, which often reflects values of normality and intentionality [2]. We leverage this information in our reasoning step annotations where available; when such cues are missing, they tend to be straightforward for humans to infer. Furthermore, once a causal model is defined, the values of sufficiency, necessity, and actual causes are deterministically derived. Regarding our synthesized data, all generated samples undergo both automated and manual validation to ensure alignment with the logic of their seed samples (Appendix A.6.3). In summary, we argue that AC-Bench is a reliable benchmark.
>
> 2. **AC-Bench vs. BBH-CJ: Measurable Improvements.** The original BBH-CJ dataset contains 187 samples. During data cleaning, we *removed 54 samples* (46 of which were samples about intention) and *corrected 2 partially flawed samples*, resulting in a cleaned dataset of 133 samples. For initial augmentation, we created 30 new samples targeting causal events that are not queried within existing stories, expanding the dataset to 163 samples. In the generation-based augmentation stage, we used 155 out of 163 samples as "seeds" and generated 6 new samples per seed, yielding 930 generated samples. Combined with the original samples, AC-Bench comprises a total of 1093 samples.
>
> 3. **AC-Bench vs. BBH-CJ:  Human & LLM Performance.** Due to time constraints and the high cognitive load of the task, we randomly selected 100 samples from AC-Bench and conducted a human evaluation. Four participants were involved, achieving accuracies of 60%, 66%, 68%, and 69%, with an average accuracy of 65.75%. This accuracy is lower than the average human performance on BBH-CJ, and the highest human performace on BBH-CJ is 100%. The result is consistent with our findings in Appendix A.4.4, which show that AC-Bench is more challenging than BBH-CJ in two key aspects: 1) More spurious correlations and fewer explicit causal cues are present in AC-Bench, which increases task complexity for both LLMs and humans [3-4]. 2) Samples querying intention are excluded from AC-Bench. As shown in Table 2 (Acc (C.) vs. Acc (I.)), these samples are generally easier because they require reasoning only over the intention factor $f_i$. Therefore, it is expected and reasonable to observe a performance drop for both LLMs and humans when shifting from BBH-CJ to AC-Bench.
>
> 4. **Performance of AC-Reason on Seed vs. Generated Subsets.** As shown in the table below, LLMs w/ AC-Reason consistently perform worse on the generated subset compared to the seed subset. This aligns with our earlier discussion that AC-Bench is more challenging due to more spurious correlations and fewer explicit causal cues.
>
>    | Model (w/ AC-Reason) | Seed Acc. | Gen Acc. |
>    | -------------------- | --------- | -------- |
>    | Qwen2.5-32B-Instruct | 71.17%    | 63.66%   |
>    | Qwen2.5-72B-Instruct | 74.85%    | 66.24%   |
>    | DeepSeek-V3          | 70.55%    | 67.10%   |
>    | Gemini-2.0-Flash     | 66.26%    | 64.73%   |
>    | Claude-3.5-Sonnet    | 75.46%    | 69.68%   |
>    | GPT-4-0613           | 76.69%    | 70.97%   |
>    | GPT-4o-2024-11-20    | 69.33%    | 67.85%   |
>
> 5. **Statistical Significance of AC-Reason on BBH-CJ.** We have conducted an approximate randomization test to assess the statistical significance of AC-Reason on BBH-CJ. For each model, we evaluated AC-Reason and three baselines across 10 runs. Each approximate randomization test was repeated 30 times with 10000 trials per run to compute the mean $p$-value. A Bonferroni correction was applied for multiple comparisons (18 in total: 6 models × 3 baselines), yielding a corrected significance threshold of 0.05/18≈0.00278. As shown below, AC-Reason achieves statistical significance in 18/18 comparisons before Bonferroni correction and 16/18 comparisons after Bonferroni correction, despite the relatively small size of BBH-CJ.
>
>    | Model                | Comparison                  | Accuracy Diff | $p$-value (± std) | Stat. Significant | Stat. Significant (Bonferroni) |
>    | -------------------- | --------------------------- | ------------- | ----------------- | ----------------- | ------------------------------ |
>    | Qwen2.5-32B-Instruct | AC-Reason vs. Vanilla       | 0.0369        | 0.0020 ± 0.0004   | √                 | √                              |
>    |                      | AC-Reason vs. Zero-shot CoT | 0.0765        | 0.0021 ± 0.0004   | √                 | √                              |
>    |                      | AC-Reason vs. Manual CoT    | 0.0353        | 0.0022 ± 0.0004   | √                 | √                              |
>    | Qwen2.5-72B-Instruct | AC-Reason vs. Vanilla       | 0.0465        | 0.0022 ± 0.0004   | √                 | √                              |
>    |                      | AC-Reason vs. Zero-shot CoT | 0.0754        | 0.0022 ± 0.0003   | √                 | √                              |
>    |                      | AC-Reason vs. Manual CoT    | 0.0642        | 0.0020 ± 0.0004   | √                 | √                              |
>    | DeepSeek-V3          | AC-Reason vs. Vanilla       | 0.0283        | 0.0079 ± 0.0009   | √                 | ×                              |
>    |                      | AC-Reason vs. Zero-shot CoT | 0.0289        | 0.0020 ± 0.0005   | √                 | √                              |
>    |                      | AC-Reason vs. Manual CoT    | 0.0604        | 0.0020 ± 0.0003   | √                 | √                              |
>    | Claude-3.5-Sonnet    | AC-Reason vs. Vanilla       | 0.0465        | 0.0080 ± 0.0007   | √                 | ×                              |
>    |                      | AC-Reason vs. Zero-shot CoT | 0.0642        | 0.0021 ± 0.0005   | √                 | √                              |
>    |                      | AC-Reason vs. Manual CoT    | 0.0754        | 0.0020 ± 0.0004   | √                 | √                              |
>    | GPT-4-0613           | AC-Reason vs. Vanilla       | 0.0866        | 0.0021 ± 0.0004   | √                 | √                              |
>    |                      | AC-Reason vs. Zero-shot CoT | 0.0813        | 0.0020 ± 0.0004   | √                 | √                              |
>    |                      | AC-Reason vs. Manual CoT    | 0.0658        | 0.0020 ± 0.0005   | √                 | √                              |
>    | GPT-4o-2024-11-20    | AC-Reason vs. Vanilla       | 0.1176        | 0.0019 ± 0.0005   | √                 | √                              |
>    |                      | AC-Reason vs. Zero-shot CoT | 0.0834        | 0.0020 ± 0.0005   | √                 | √                              |
>    |                      | AC-Reason vs. Manual CoT    | 0.0909        | 0.0021 ± 0.0004   | √                 | √                              |
>
> 6. **Multivariate Causation.** BBH-CJ is designed to assess causality based on whether a single atomic candidate cause leads to an outcome; thus, it does not consider multivariate causation.
>
> **References**
>
> [1] Suzgun, M., Scales, N., Schärli, N., Gehrmann, S., Tay, Y., Chung, H. W., ... & Wei, J. (2023, January). Challenging BIG-Bench Tasks and Whether Chain-of-Thought Can Solve Them. In *ACL (Findings)*.
>
> [2] Srivastava, A., Kleyko, D., & Wu, Z. (2023). Beyond the imitation game: Quantifying and extrapolatingthe capabilities of language models. *Transactions on Machine Learning Research*, (5).
>
> [3] Matute, H., Blanco, F., Yarritu, I., Díaz-Lago, M., Vadillo, M. A., & Barberia, I. (2015). Illusions of causality: How they bias our everyday thinking and how they could be reduced. *Frontiers in psychology*, *6*, 888.
>
> [4] Vasilyeva, N., & Lombrozo, T. (2015). Explanations and causal judgments are differentially sensitive to covariation and mechanism information. In *Proceedings of the Annual Meeting of the Cognitive Science Society* (Vol. 37).

---

> ### Comment · Reviewer_ojMy · 2025-08-07
> **Thank you for your rebuttal.**
>
> I have raised my score by one point, because the significance results on LLM performance seem promising.
>
> I remain doubtful about the reliability of actual causation judgments on AC-BENCH, given that 1) human performance is consistently low at 66%, 2) the literature often disagrees in judgments of actual causation, and 3) labeling was not crowd-sourced and was performed by a single human - inter-annotator reliability measures cannot be calculated.

---

> > ### Author Response · Authors · 2025-08-07
> > **Thanks & Further Clarifications**
> >
> > Thanks for your response and for raising the score! We sincerely appreciate your thoughful engagement.
> >
> > Regarding your concerns about the reliability of AC-Bench, we would like to provide further clarifications:
> >
> > 1. **Low human performance does not imply low data quality.** The Yes/No labels in BBH-CJ were *not* labeled by us, but were derived from rigorous human experiments. These labels reflect *how the majority of people judge causality* in given scenarios. Naturally, achieving very high accuracy is challenging for both individuals and LLMs, as the underlying distributions vary: some cases reflect a clear consensus (e.g., ~100% vs. 0%), while others are more ambiguous (e.g., ~60% vs. 40%). This is an inherent feature of binary datasets, not a flaw in data quality. In contrast, AC-Bench labeling focuses on intermediate reasoning steps. By decomposing the causal judgment into more concrete subtasks (e.g., identifying events and inferring their factor values), the labeling becomes more straightforward and interpretable for human annotators.
> > 2. **Actual causality is just one component of the broader causal judgment process.** While actual causality aims to simulate human causal judgment within a formal framework (e.g. using structural causal models or SCMs), it can *not* (fully) capture informal influences such as normality and intention. These informal factors often lead people to judge that certain actual causes are (or are not) causes. Therefore, *the "judgment" produced by actual causality alone is insufficient, and this is precisely why actual causes are not always judged as causes in the context of causal judgment*. To bridge this gap, AC-Bench integrates both formal and informal factors, aiming to better model human causal judgment.
> >
> > We hope this additional clarification helps support a more informed evaluation of AC-Bench.

---

### Official Review · Reviewer_H9zo · 2025-07-03

**Clarity:** 3
**Significance:** 1
**Originality:** 2
**Rating:** 2
**Confidence:** 5

**Summary:**

The authors use LLMs to augment Big-Bench Hard Causal Judgment and develop a hard coded chain of thought multi-step protocol to break down problems related to judging actual causation. They evaluate a large suite of open and closed weight LLMs and carry out ablations of their protocol.

**Questions:**

1.I think there is significant confusion as to what the goal of actual causation is in this paper. Actual causation was an attempt to formalize how people actually make causal judgments. The HP definition of AC is an attempt to capture human causal judgments (although it is incomplete and has many additions related to causal selection). Thus the goal of a causal reasoning system is not to match HP but rather to match people. Congruence with HP would be a benchmark of how well LLMs can follow formal logic but that clearly isn’t the goal. Towards this issue, the authors make claim multiple times that their system exceeds human performance. However, this doesn’t really make sense. Human judgments of causal selection are the end-point not another “system” to benchmark against.

2.How is the modified definition of HP different from the original HP definitions? What is the value of the proofs of the propositions. These looks very similar to the original and its not clear if there is novelty or value here.   3.The stimuli in AC-bench are highly stylized situations taken from the psychology literature that are designed to study and isolate the specific factors chosen by the authors. These are not naturalistic or realistic situations where judging AC correctly would be of any use. How does AC-reason fare on more naturalistic stimuli and how well does it approximate human reasoning. This would require a new benchmark and the collection of new human data.

**Ethical Concerns:**

["NO or VERY MINOR ethics concerns only"]

**Final Justification:**

I have read the other reviews and have decided to keep my score

**Limitations:**

yes

**Quality:**

2

**Strengths And Weaknesses:**

Strengths:
+ Many LLMs models are tested which shows some degree of robustness
+ Some amount of manual data curation to clean up Big-Bench Hard Causal Judgment
+ Nicely written introduction. Paper is mostly clearly written
+ Evaluation of causal reasoning in LLMs is an important topic
+ 1/3 of the dataset is manually verified

Weaknesses:
-Confusion as to what actual causation is actually about. (See questions)
-The difference between AC-bench and Big-Bench Hard Causal Judgment is just LLM augmentation with GPT 4o.
-AC-bench is not shown to actually be more valuable for evaluation than the original dataset.
-Comparison with humans doesn’t make sense (See questions)
-No evaluation of AC-reason on naturalistic data that is outside the narrow scope of the highly stylized psychology experiments studied in AC-bench.
-Presentation of the theoretical results makes it seem like they are novel. They are not.

I believe this is at best a minor contribution and is more likely to add confusion to the literature on causation in LLMs than to advance it.

---

> ### Author Rebuttal · Authors · 2025-07-31
>
> **Goal of Actual Causation.** We would like to clarify a factual misunderstanding. *The "match people" goal is precisely what our work aims to achieve.* Our task is causal judgment, and the dataset we use is the Causal Judgment subset of BIG-Bench Hard [1], originally derived from BIG-Bench [2]. Each story in Big-Bench is grounded in rigorous human experiments that produce ground-truth causal judgments [2]. Therefore, the goal of the task is to "match people." Regarding the method, the HP definition is one of several causal factors incorporated in AC-Reason. It constitutes part of the causal judgment process and formalizes how people naturally attribute causes and assigns responsibility (as also acknowledged in your comment), which is precisely why we integrate it into our framework. In short, actual causation, as applied as a formalization tool in our work, does aim to "match people."
>
> **Comparison with Humans.** In claiming that our system outperforms the average human rater, our aim is to demonstrate that AC-Reason can effectively and efficiently replicate human causal judgments grounded in rigorous human experiments [2], without relying on domain experts or crowd annotators. This highlights the practicality and scalability of AC-Reason for automated actual causality analysis in large-scale applications. It is also important to clarify that the human performance is taken directly from prior work [1-2], and we follow standard practice by comparing against it.
>
> **Naturalistic/Realistic Data.** Collecting fully naturalistic data is outside the scope of this paper and not the focus of our contribution. To the best of our knowledge, no such open-source dataset currently exists. While BBH-CJ (from which AC-Bench is derived) does not consist of fully naturalistic narratives, it is nonetheless a rigorously constructed dataset, based on controlled human experiments [2]. It includes real-world factors related to causal judgment, such as morality, normality, and intent [3], and thus widely adopted in recent works as a reliable benchmark for evaluating actual causality [1-2, 4-5]. In this context, identifying actual causes is indeed essential. Just as humans typically begin by contributing (or partial) causes, and then reason further based on influencing factors such as morality, normality, and intent [6-7], AC-Reason follows a similar multi-step reasoning structure.
>
> **AC-Bench vs. BBH-CJ.** *AC-Bench is not a simple augmentation of BBH-CJ with GPT-4o* for the following reasons: 1) We carefully inspected and cleaned the entire BBH-CJ dataset. 2) AC-Bench includes annotated reasoning steps, which are absent in BBH-CJ. 3) The first stage of augmentation is not based on GPT-4o. Instead, we manually crafted additional causal queries based on unasked causal events in existing stories. 4) The generated samples via GPT-4o undergo both automated and manual validation. The construction of AC-Bench is non-trivial, which involves high cognitive load. Moreover, *AC-Bench offers greater evaluation value than BBH-CJ* since: 1) It is focused solely on causation queries, improving task specificity. 2) It is more challenging, as shown in our analysis (Appendix A.4.4). The generated samples contain more spurious correlations and fewer explicit causal cues, increasing task complexity and better testing generalization.
>
> **HP Definitions & Propositions.** We have carefully cited Pearl's book [3] as the source of the original, updated, and modified versions of the HP definition. Regarding Proposition 1 [3], its role is to clarify the relationship between necessary causes and actual causes, which helps us design Algorithm 1 and label the reasoning steps.
>
> **References**
>
> [1] Suzgun, M., Scales, N., Schärli, N., Gehrmann, S., Tay, Y., Chung, H. W., ... & Wei, J. (2023, January). Challenging BIG-Bench Tasks and Whether Chain-of-Thought Can Solve Them. In *ACL (Findings)*.
>
> [2] Srivastava, A., Kleyko, D., & Wu, Z. (2023). Beyond the imitation game: Quantifying and extrapolatingthe capabilities of language models. *Transactions on Machine Learning Research*, (5).
>
> [3] Sloman, S. A., & Lagnado, D. (2015). Causality in thought. Annual review of psychology, 66(1), 223-247.
>
> [4] Kiciman, E., Ness, R., Sharma, A., & Tan, C. (2023). Causal reasoning and large language models: Opening a new frontier for causality. *Transactions on Machine Learning Research*.
>
> [5] Chen, S., Peng, B., Chen, M., Wang, R., Xu, M., Zeng, X., ... & Lu, C. (2024). Causal evaluation of language models. arXiv preprint arXiv:2405.00622.
>
> [6] Henne, P., O’Neill, K., Bello, P., Khemlani, S., & De Brigard, F. (2021). Norms affect prospective causal judgments. *Cognitive Science*, *45*(1), e12931.
>
> [7] Cushman, F. (2008). Crime and punishment: Distinguishing the roles of causal and intentional analyses in moral judgment. *Cognition*, *108*(2), 353-380.

---

> > ### Comment · Reviewer_H9zo · 2025-08-01
> >
> > Thank you for the response. I have read your rebuttal, but have decided to keep my score unchanged.

---

### Official Review · Reviewer_piX2 · 2025-07-03

**Clarity:** 3
**Significance:** 3
**Originality:** 2
**Rating:** 5
**Confidence:** 2

**Summary:**

The paper focuses on actual causality: event based word-problems. To evaluate actual causality, the authors build upon big bench hard-causal judgment that has manually annotated reasoning steps for each query to generate AC-BENCH. They also propose AC-REASON that integrates AC theory into the reasoning mechanism for LLMs.  AC-REASON involves three stage processing: Causal setting establishment, causal factor analysis and finally AC reasoning. With AC-REASON, the authors show significant improvement over non-reasoning models particularly.

**Questions:**

- What is the influence of each stage on the final performance?

- What is the realistic application where AC can be used?

- Minor: the blue and orage colours are difficult to read, could you make it more legible please?

**Ethical Concerns:**

["NO or VERY MINOR ethics concerns only"]

**Limitations:**

Limitations not mentioned

**Quality:**

3

**Strengths And Weaknesses:**

# Strengths

- There seem to be less work in actual causality with LLMs so this benchmark seems to be a good contribution in the field.

- The authors perform extensive experiments with their proposed method.

- The appendix is quite informative with detailed results.

- AC CoT is guided in AC theory.

- Prompts were provided in the appendix, that gives a qualitative comparison to the reader.

# Weakness

- AC-REASON is quite a small benchmark with only ~1k samples.

- I suppose AC-REASON is computationally expensive, discussion around that would be useful.

- Additionally the emperical influence of each of the stage vs computation could be useful to motivate?

---

> ### Author Rebuttal · Authors · 2025-07-30
>
> 1. **Size of AC-Bench.** AC-Bench contains only ~1K samples, as *collecting actual causality data is inherently challenging*. Big-Bench Hard [1] is derived from Big-Bench [2], which collected stories from ~30 papers published between 1989 and 2021. From this extensive effort, only ~190 samples were gathered for the Causal Judgment subset. In Big-Bench Hard, the corresponding subset contains 187 samples, among which only 141 focus specifically on causation. Moreover, *annotating and inspecting actual causality data imposes a high cognitive load*. For example, during data cleaning on Big-Bench Hard Causal Judgment, we performed a full dataset inspection. During annotation, all events must be labeled in accordance with a causal model, and values for each causal factor must be carefully assigned. After data generation, we further inspect 1/3 of the dataset with reasoning steps. Considering these labor-intensive requirements, we argue that a dataset of ~1K samples offers a practical balance between data size and quality.
>
> 2. **Influence of Each Stage on Final Performance.** The influence of each stage in AC-Reason is quantitatively reflected in the ablation study results provided in Table 4. Beyond this, we conducted both factor and causal analyses to evaluate how correctly inferring individual causal factor values affects final performance. The results of these analyses are illustrated in Figures 3 and 4.
>
> 3. **Computational Costs & Costs by Stage.** We report computational costs of applying AC-Reason across various LLMs on BBH-CJ. As for *time overhead*, we present average runtimes per stage for samples about causation and overall averages across all samples. As for *memory overhead*, we report estimated memory usage for open-source LLMs, which primarily depends on model parameters and KV cache. Importantly, we note that *analyzing runtime stage-by-stage offers limited insight*, as performace gains primarily emerge in the third stage. The first two stages serve mainly to extract structured causal knowledge (i.e. the causal setting and values of causal factors), while the third stage is where this knowledge is operationalized via algorithmic reasoning.
>
>    | Model                | Vanilla Acc. | Stage 1 Acc. | Stage 1 Runtime  Avg. | Stage 1+2 Acc. | Stage 2 Runtime Avg. | Stage 1+2+3 Acc. | Stage 3 Runtime Avg. | Overall Avg Runtime | Estimated Memory Usage |
>    | -------------------- | ------------ | ------------ | --------------------- | -------------- | -------------------- | ---------------- | -------------------- | ------------------- | ---------------------- |
>    | Qwen2.5-32B-Instruct | 68.98%       | 65.82%       | 5.96s                 | 70.45%         | 6.45s                | 72.55%           | 9.87s                | 14.16s              | ~64.5GB                |
>    | Qwen2.5-72B-Instruct | 68.98%       | 67.73%       | 5.40s                 | 67.84%         | 6.19s                | 73.62%           | 12.02s               | 15.34s              | ~144.6GB               |
>    | DeepSeek-V3          | 69.70%       | 70.57%       | 3.21s                 | 68.66%         | 3.19s                | 73.44%           | 3.98s                | 7.28s               | ~102.7GB               |
>    | Claude-3.5-Sonnet    | 66.49%       | 59.39%       | 2.61s                 | 60.01%         | 2.98s                | 74.33%           | 4.64s                | 6.64s               | 0GB                    |
>    | GPT-4o-2024-11-20    | 62.03%       | 57.45%       | 1.66s                 | 60.28%         | 1.81s                | 73.98%           | 1.75s                | 3.97s               | 0GB                    |
>    | GPT-4-0613           | 67.38%       | 63.12%       | 4.79s                 | 64.78%         | 5.49s                | 75.04%           | 3.94s                | 10.98s              | 0GB                    |
>
> 4. **Applications of Actual Causality.** Actual causality has a wide range of applications across both theoretical and practical domains. In the legal field, it underpins legal reasoning [3] and the attribution of legal responsibility [4]. It is also vital in technical contexts such as machine failure diagnosis [5], root cause analysis in configurable systems [6-7], and explainable AI [8-9].
>
> 5. **Visualizations.** We appreciate the suggestion to improve visual representations. While the rebuttal phase supports text-only responses, we will revise and enhance the visualization in the final version of the paper.
>
>
> **References**
>
> [1] Suzgun, M., Scales, N., Schärli, N., Gehrmann, S., Tay, Y., Chung, H. W., ... & Wei, J. (2023, January). Challenging BIG-Bench Tasks and Whether Chain-of-Thought Can Solve Them. In *ACL (Findings)*.
>
> [2] Srivastava, A., Kleyko, D., & Wu, Z. (2023). Beyond the imitation game: Quantifying and extrapolatingthe capabilities of language models. *Transactions on Machine Learning Research*, (5).
>
> [3] Lagnado, D. A., & Gerstenberg, T. (2017). Causation in legal and moral reasoning.
>
> [4] Andreas, H., Armgardt, M., & Gunther, M. (2023). Counterfactuals for causal responsibility in legal contexts. *Artificial intelligence* *and law*, *31*(1), 115-132.
>
> [5] Lu, P., Ruchkin, I., Cleaveland, M., Sokolsky, O., & Lee, I. (2023, June). Causal repair of learning-enabled cyber-physical systems. In *2023 IEEE International Conference on Assured Autonomy (ICAA)* (pp. 1-10). IEEE.
>
> [6] Dubslaff, C., Weis, K., Baier, C., & Apel, S. (2022, May). Causality in configurable software systems. In *Proceedings of the 44th International Conference on Software Engineering* (pp. 325-337).
>
> [7] Sharma, A., Li, H., & Jiao, J. The Counterfactual-Shapley Value: Attributing Change in System Metrics. In *NeurIPS 2022 Workshop on Causality for Real-world Impact*.
>
> [8] Blake, N., Kelly, D. A., Chanchal, A., Kapllani-Mucaj, S., Thomas, G., & Chockler, H. (2025). SpecReX: Explainable AI for Raman Spectroscopy. *arXiv preprint arXiv:2503.14567*.
>
> [9] Chockler, H., Kelly, D. A., Kroening, D., & Sun, Y. (2024). Causal explanations for image classifiers. *arXiv preprint arXiv:2411.08875*.

---

> > ### Comment · Reviewer_piX2 · 2025-08-02
> >
> > I would like to thank the authors for pointing out the relevant parts of the paper and conducting additional experiments. My judgement of the paper remains positive with my initial score.

---

### Official Review · Reviewer_LoDP · 2025-07-07

**Clarity:** 3
**Significance:** 2
**Originality:** 3
**Rating:** 3
**Confidence:** 4

**Summary:**

This paper presents an LLM-based reasoning framework, AC-Reason, for addressing actual causality (AC) tasks, which require identifying formal causal factors between individual events. The proposed framework integrates LLMs with AC theory and enhances reasoning performance on AC-Bench, a new benchmark focusing on AC relationships.

**Questions:**

Refer to the weaknesses mentioned above. Below are questions that may help improve the revision of this paper.

- Determining causal formulas in AC queries can be understood in the same context as inferring symbolic representations. When compared to recent research that generates logic skeletons for reasoning (e.g., ProverGen), what are the technical advantages of the template-based framework proposed in this study? I understand that empirical comparison would be difficult due to the absence of benchmark results, it seems to be a necessary comparison that should be presented in this paper.
- The proposed AC-reason framework appears to provide answers to AC queries about specific causal events. When determining which causal events correspond to actual causes, by what criteria can we verify whether the causes inferred by LLMs based on a given story are robust? For example, in Figure 1, both Kate's and Janet's actions correspond to actual causes of the machine breaking, but depending on hidden contexts that cannot be identified from the story (e.g., it is not Kate's responsibility, so there's no obligation), Kate's action might be considered a fragile (or uncertain) cause.

[ProverGen] Qi et al., Large Language Models meet symbolic provers for logical reasoning evaluation, *ICLR*, 2025.

**Ethical Concerns:**

["NO or VERY MINOR ethics concerns only"]

**Final Justification:**

I listed some concerns at the following link.
- Link: https://openreview.net/forum?id=EXChZFno6e&noteId=uTkjGedMb4

Having re-examined their arguments and the manuscript, while the rebuttal clarifies several aspects of the work, it remains some concerns.

- Technical Contribution of AC-Reason.

I accept that Algorithm 1 is the core contribution. However, its novelty should be carefully qualified. The algorithm is an explicit, rule-based implementation of established principles from the causal judgment literature. The contribution is therefore less about creating new causal theory and more about the skillful codification and integration of existing theory into a computational framework. While this is a valuable application-focused contribution, its theoretical novelty is limited. Furthermore, the rebuttal states that "we incorporate self-reflection and self-verification mechanisms by using slow thinking models." This point seems overstated. Based on the ablation study (Table 4), this appears to be a case of running the entire AC-REASON pipeline on more advanced models, rather than an integrated, iterative self-verification step within the algorithm itself. The core framework does not seem to possess an intrinsic reflection mechanism. The reliance on the quality of the base LLM for the initial, crucial step of factor extraction remains a significant limitation.

- On the Comparison with Symbolic Methods.

The authors compellingly argue that a direct comparison is not meaningful due to the necessity of integrating informal factors (normality, intention) that are outside the scope of purely logical systems. However, the authors may be too quick to dismiss the relevance of symbolic methods. They concede that a tool like ProverGen could be useful for the actual cause subtask but frame this as a "minor part" of the pipeline. I question whether this part is truly "minor." The $f_ac$ factor, grounded in the formal HP definition, is arguably the most complex and central formal component of their framework. The paper's own analysis (Figure 3) shows that many LLMs struggle to use this factor correctly, suggesting it is a critical bottleneck. Therefore, the potential for a hybrid approach that uses a symbolic prover for this specific, challenging step seems like a significant and underexplored avenue that warrants more than a brief dismissal.

- On the AC-BENCH Contribution.

The process of the benchmark's creation deserves careful examination. The data augmentation strategy involved rewriting stories while explicitly keeping the "reasoning logic" and "factor values unchanged." This creates a risk that the benchmark is inadvertently structured in a way that is perfectly tailored to the AC-REASON framework. A method designed to extract these specific factors and plug them into a rule-based system is likely to perform well on a dataset where the narrative was generated to fit those exact factors. This raises a potential question about how well the framework's performance on AC-BENCH would generalize to real-world causal scenarios that have not been curated in this manner.

While the above concerns remain, the paper has many interesting parts. I raise my rating (2 → 3).

**Limitations:**

Yes.

**Paper Formatting Concerns:**

There is no issue.

**Quality:**

2

**Strengths And Weaknesses:**

**Strengths**

- (Quality & Originality) Based on AC-theory, the AC-Reason framework proposed in this paper shows improved performance in the pilot study and the presented AC-Bench when applied to various LLMs (Qwen, DeepSeek, Claude, GPT-4o), contrary to previous naive Chain-of-Thought methods.
- (Clarity) The paper is written very clearly, and presents technical content in a way that is not difficult to read.

**Weaknesses**

The reviewer would like to point out the following issues:

- In Section 3.1, the Causal Setting Establishment part defines the causal model $M$, which is a crucial element in determining the actual cause and sufficient cause in Definitions 1 and 2. However, the fact that the method for defining this part relies on handcrafted templates and LLM outputs mean that the proposed technique is subject to the same limitations as existing prompt engineering or CoT techniques. Consequently, from the reviewer's perspective, it is uncertain to find clear technical contributions of the proposed framework compared to previous prompt engineering approaches to causal reasoning.
- The contribution of AC-Bench construction proposed in this paper seems more appropriate to be evaluated in the Datasets & Benchmarks Track rather than the Main Track. Although contributions regarding new datasets should rightfully be evaluated in the Main Track as well, when primarily assessing technical contributions or scientific insights, it is unconvincing to judge that the contribution of the AC-reason framework presented in this paper is sufficient by NeurIPS standards.

---

> ### Author Rebuttal · Authors · 2025-07-31
>
> 1. **Limitations of AC-Reason vs. Prompting-Based Baselines.** While the first two steps of AC-Reason do leverage the internal knowledge of LLMs, we would like to emphasize that: 1) These steps are inherently challenging to perform using external tools, and often time-consuming for human experts or crowd annotators in applied settings. Prior work such as COAT [1], though not focusing on actual causality, adopts a similar pipeline where LLMs are used to identify variables in the causal model and causal discovery algorithms are applied subsequently. The promising results from COAT support the feasibility and effectiveness of such a hybrid pipeline. 2) In the domain of actual causality, the scarcity of high-quality data presents a key challenge. For instance, the Big-Bench Hard Causal Judgment dataset contains only 187 samples, making it impractical to rely on post-training [2]. To compensate for the inherent limitations of prompting, we incorporate self-reflection and self-verification mechanisms by using slow thinking models such as DeepSeek-R1and QwQ-32B. As shown in our ablation study (Table 4), these models yield substantial performance improvements in the third steps of AC-Reason, indicating that the first two steps effectively extract better causal knowlege, i.e., the causal setting and values of causal factors.
> 2. **Technical Contribution of AC-Reason.** The core technical contribution of AC-Reason lies in its *synergistic combination of LLM capabilities and causal theory*, wich each component applied to the task for which it is best suited. Specifically, 1) LLMs are used to establish the causal setting and infer the values of causal factors, tasks at which they excel [3]; 2) the theoretical foundation of actual causality (via the HP definition) and causal judgment (via factors such as normality and intention) is used to guide a transparent and interpretable reasoning process (via Algorithm 1), leading to rigorous reasoning steps and explanations. This design in non-trivial, especially as formalized in Algorithm 1, and reflects a deliberate application of theoretical insights to actual causality reasoning in practice. The overall pipeline is also aligned with recent work such as COAT [1], reinforcing its validity. In this light, AC-Reason meets the expectations of the Applications Track, offering both practical utility and theoretical grounding.
> 3. **Comparison with ProverGen.** We argue that AC-Reason is not meaningfully comparable to ProverGen, due to fundamental differences in task formulation and reasoning paradigms. From a broader perspective, actual causality requires integrating both formal and informal factors. Formal factors include sufficiency, necessity, and actual causes, all of which are grounded in the semantics of structural causal models (SCMs). Informal factors, such as normality and intention, are not part of the SCM itself, but are essential to causal judgment, and are grounded in empirical findings from rigorous human experiments [4]. However, these information can not be adequately represented by logical skeletons or purely symbolic entailment. From a narrower perspective, only one component of AC-Reason (the actual cause factor $f_{ac}$) involves reasoning over causal formulas (though our method does not explicitly perform this step) [5]. In this specific subtask, a tool like ProverGen could potentially be useful. However, this represents only a minor part of the overall AC-Reason pipeline. Therefore, a direct comparison between AC-Reason and ProverGen is not meaningful.
> 4. **Verification of Robust Causes.** We would first like to clarify the distinction between "actual causes" and "causes" as used in our paper, and how this relates to the goal of causal judgment. In our paper, "actual causes" refer to partial causes as defined by the modified HP definition [5]. In contrast, "causes" in the context of causal judgment refer to the outcomes of human attributions of responsibility. Therefore, judging "causes" involves (but is not limited to) consideration of "actual causes". For example, if both $X_1$ and $X_2$ jointly cause $Y$, they are both actual causes by Definition 1. However, humans rarely attribute $Y$ to $X_1$ in isolation; the attribution is often made to the combination of $X_1$ and $X_2$. Regarding the example in Figure 1, we would like to clarify several factual misunderstandings: 1) The statement that "It is not Kate's responsibility" is not a hidden context. This information is explicitly stated in the story: "If Janet does not put oil in the machines, it is not Kate's responsibility to do so." In AC-Reason, this is captured by normality ($f_n$ takes on False in this case), indicating that Kate did not violate any norm, e.g., factory policy or instruction from her mentor. 2) Kate's behavior is an actual cause (a partial cause) of the outcome, and a lack of obligation does not dequalify it as a cause. Whether Kate's behavior is judged as a cause in the context of causal judgment depends on her intention ($f_i$ takes on True in this case). The story notes: "Kate noticed that Janet did not put oil in the machine, and Kate also did not put oil in the machine." The intentional inaction increases the causal strength (or contribution) of Kate's behavior, making it judged as a cause. This is captured by Line 16 of Algorithm 1. 3) The "fragile" or "uncertain" causes you refer to stem from the binary nature of causal decisions in the BBH-CJ dataset. However, AC-Reason supports and reasons through such binary causal decisions using Algorithm 1. Finally, the verification of "actual causes" in the second step of AC-Reason can leverage Proposition 1, which states that necessary causes are a subset of actual causes. Since LLMs demonstrate high accuracy in inferring necessity (as shown in Table 3 and consistent with prior work [3]), this provides a reliable basis for this verification. Additional verification can be achieved through self-reflection mechanisms, such as employing "slow thinking" LLMs as backbones, as discussed earlier.
>
> **References**
>
> [1] Liu, C., Chen, Y., Liu, T., Gong, M., Cheng, J., Han, B., & Zhang, K. (2024). Discovery of the hidden world with large language models. *Advances in Neural Information Processing Systems*, *37*, 102307-102365.
>
> [2] Suzgun, M., Scales, N., Schärli, N., Gehrmann, S., Tay, Y., Chung, H. W., ... & Wei, J. (2023, January). Challenging BIG-Bench Tasks and Whether Chain-of-Thought Can Solve Them. In *ACL (Findings)*.
>
> [3] Kiciman, E., Ness, R., Sharma, A., & Tan, C. (2023). Causal reasoning and large language models: Opening a new frontier for causality. *Transactions on Machine Learning Research*.
>
> [4] Srivastava, A., Kleyko, D., & Wu, Z. (2023). Beyond the imitation game: Quantifying and extrapolatingthe capabilities of language models. *Transactions on Machine Learning Research*, (5).
>
> [5] Halpern, J. Y. (2016). *Actual causality*. MiT Press.

---

> > ### Comment · Reviewer_LoDP · 2025-08-06
> > **Re-evaluation**
> >
> > I appreciate the authors for their detailed rebuttal. Having re-examined their arguments and the manuscript, while the rebuttal clarifies several aspects of the work, it remains some concerns.
> >
> > - **Technical Contribution of AC-Reason.**
> > I accept that Algorithm 1 is the core contribution. However, its novelty should be carefully qualified. The algorithm is an explicit, rule-based implementation of established principles from the causal judgment literature. The contribution is therefore less about creating new causal theory and more about the skillful codification and integration of existing theory into a computational framework. While this is a valuable application-focused contribution, its theoretical novelty is limited. Furthermore, the rebuttal states that "we incorporate self-reflection and self-verification mechanisms by using slow thinking models." This point seems overstated. Based on the ablation study (Table 4), this appears to be a case of running the entire AC-REASON pipeline on more advanced models, rather than an integrated, iterative self-verification step within the algorithm itself. The core framework does not seem to possess an intrinsic reflection mechanism. The reliance on the quality of the base LLM for the initial, crucial step of factor extraction remains a significant limitation.
> >
> > - **On the Comparison with Symbolic Methods.**
> > The authors compellingly argue that a direct comparison is not meaningful due to the necessity of integrating informal factors (normality, intention) that are outside the scope of purely logical systems. However, the authors may be too quick to dismiss the relevance of symbolic methods. They concede that a tool like ProverGen could be useful for the actual cause subtask but frame this as a "minor part" of the pipeline. I question whether this part is truly "minor." The $f_ac$ factor, grounded in the formal HP definition, is arguably the most complex and central formal component of their framework. The paper's own analysis (Figure 3) shows that many LLMs struggle to use this factor correctly, suggesting it is a critical bottleneck. Therefore, the potential for a hybrid approach that uses a symbolic prover for this specific, challenging step seems like a significant and underexplored avenue that warrants more than a brief dismissal.
> >
> > - **On the AC-BENCH Contribution.**
> > The process of the benchmark's creation deserves careful examination. The data augmentation strategy involved rewriting stories while explicitly keeping the "reasoning logic" and "factor values unchanged." This creates a risk that the benchmark is inadvertently structured in a way that is perfectly tailored to the AC-REASON framework. A method designed to extract these specific factors and plug them into a rule-based system is likely to perform well on a dataset where the narrative was generated to fit those exact factors. This raises a potential question about how well the framework's performance on AC-BENCH would generalize to real-world causal scenarios that have not been curated in this manner.
> >
> > While the above concerns remain, the paper has many interesting parts. I raise my rating (2 → 3). I strongly urge the authors to incorporate a discussion of these remaining concerns in their revised version.

---

> > > ### Author Response · Authors · 2025-08-08
> > > **Thanks & Further Clarifications**
> > >
> > > Thanks for your detailed response and for raising the score!
> > >
> > > We have carefully considered the remaining concerns and find them insightful. Below, we provide additional results and clarifications that we hope will (at least partly) address these points before revising our paper.
> > >
> > > **1. AC-Reason: Reflection and Verification.**
> > >
> > > We agree that simply employing a stronger backbone is insufficient to address issues of self-reflection and self-verification. To this end, we propose two mitigation strategies:
> > >
> > > - **Verification via Proposition 1.** We applied Proposition 1 to our existing results and found that *all samples passed the verification test*. That is, all causal events inferred as necessary causes were also inferred as actual causes. Therefore, this verification is not needed.
> > > - **Integrating Reflection and Verification Steps.** We are trying to incorporate explicit self-reflection and verification steps after the first two steps of AC-Reason. Due to time constraints in the discussion phase, these results will be reflected in the revised version of the paper.
> > >
> > > **2. AC-Reason & Symbolic Methods.**
> > >
> > > We argue that $f_{ac}$ can not be solved using FOL-based methods such as ProverGen, and that unfaithful reasoning does not stem from low $f_{ac}$ accuracy, but from coordination failures across multiple factors.
> > >
> > > - **FOL Can Not Express Counterfactuals.** In the previous rebuttal, we mistakenly stated that "$f_{ac}$ can be inferred using ProverGen." This is incorrect. The HP definition is fundamentally counterfactual-based, requiring *counterfactual reasoning* within structural causal models (SCMs). FOL can not express such counterfactual semantics. For example, the counterfactual "If Alice had called, Bob would not have come" is formalized as $do(A:=\text{false})\rightarrow \lnot B$, where $A$ and $B$ represent "Alice had called" and "Bob had come," respectively. This expression requires modeling the semantics of SCMs (typically causal graphs) and interventions (which involves operating on causal graphs using do-operator), which can not be captured by FOL [1-2]. Therefore, FOL-based methods like ProverGen can not infer $f_{ac}$.
> > > - **Unfaithful Reasoning is Not Due to Low $f_{ac}$ Accuracy.** As shown in Table 3, most LLMs achieve over 70% accuracy when predicting $f_{ac}$ independently. Thus, failures in faithful reasoning are actually attributed to the model's inability to jointly and correctly infer $f_{ac}$ alongside other factors. Even when $f_{ac}$ is correct, errors in other factors can lead Algorithm 1 down an incorrect reasoning path. Therefore, the core challenge is not the isolated accuracy $f_{ac}$, but rather the failure of coordinated multi-factor inference.
> > >
> > > In summary, comparisons between AC-Reason and ProverGen are not meaningful. While more expressive symbolic methods (e.g., higher-order logics or specialized methods) may offer future directions, they lie beyond the scope of this work. Our priority should be improving the joint inference of all causal factors of a causal event rather than optimizing $f_{ac}$ alone.
> > >
> > > **3. AC-Bench: Realistic Simulation.**
> > >
> > > While fully real-world causal scenarios are beyond the scope of this work and remain a promising direction for future research, we argue that *AC-Bench provides a more realistic simulation than BBH-CJ*. Although it retains the same "reasoning logic" and "factor values" as BBH-CJ, the generation pipeline introduces additional complexity by adding and removing details to adapt to the new story setting. These modifications result in samples with *more spurious correlations* and *less explicit causal cues* (e.g., statements like "$X_1$ and $X_2$ jointly cause $Y$"), as detailed in Appendix A.4.4. In real-world contexts, such spurious correlations are common, and people often mention possible causes without clearly articulating explicit causal cues [3-4]. The presence of these features significantly influences 1) human causal judgment and 2) how LLMs set the causal frame and infer factor values.
> > >
> > > **References**
> > >
> > > [1] Thielscher, M. (1999, September). A theory of first-order counterfactual reasoning. In *Annual Conference on Artificial Intelligence* (pp. 137-148). Berlin, Heidelberg: Springer Berlin Heidelberg.
> > >
> > > [2] Bochman, A., & Lifschitz, V. (2015, February). Pearl's causality in a logical setting. In *Proceedings of the AAAI Conference on Artificial Intelligence* (Vol. 29, No. 1).
> > >
> > > [3] Matute, H., Blanco, F., Yarritu, I., Díaz-Lago, M., Vadillo, M. A., & Barberia, I. (2015). Illusions of causality: How they bias our everyday thinking and how they could be reduced. *Frontiers in psychology*, *6*, 888.
> > >
> > > [4] Vasilyeva, N., & Lombrozo, T. (2015). Explanations and causal judgments are differentially sensitive to covariation and mechanism information. In *Proceedings of the Annual Meeting of the Cognitive Science Society* (Vol. 37).

---

### Decision · Program_Chairs · 2025-09-17

**Decision:**

Reject

**Comment:**

This paper introduces a benchmark for actual causality and proposes a method that combines LLMs with an implementation of formal AC definition.
Strength: A benchmark on actual causality is very welcome! Very few exist, and this can be useful to the field
Weakness: the AC-Reason framework is a straightforward implementation of known definitions, paper needs conceptual clarity in many places

While reviewers appreciated the benchmark contribution, I tend to agree with Reviewer LoDP  that this paper would have fit better in the datasets track (and unfortunately, we cannot transfer it now). At the same time, the paper would also benefit from conceptual clarity. While the dataset is a great contribution, some of the experiments, e.g., comparing AC-reason to an average human rater baseline are confusing because the ground-truth of the dataset is also extracted from (majority of) humans. The paper does have a lot of merits, but I believe it needs one round of revision before it is ready. As one suggestion to help clarity, I suggest the authors to separate informal factors and formal HP definitions and define where the method's contribution is coming from. I suspect it is coming from adding informal factors such as normality through LLM, and that can be highlighted more clearly.